# Machine learning-led semi-automated medium optimization reveals salt as key for flaviolin production in *Pseudomonas putida*

Apostolos Zournas [1,2,3,8], Matthew R. Incha[1,2,3,8], Tijana Radivojevic[1,2,3], Vincent Blay [1,3], Jose Manuel Martí [1,2,3], Zak Costello[1,2,3], Matthias Schmidt [1,3], Tan Chung[1,3,4], Mitchell G. Thompson [1,3], Allison Pearson[1,3], Patrick C. Kinnunen [1,2,3], Thomas Eng [1,3], Christopher E. Lawson[1,3], Stephen Tan[1,2,3], Tadeusz Ogorzalek[1,2,3], Nurgul Kaplan[1,2,3], Mark Forrer[2,3,5], Tyler Backman [1,3], Aindrila Mukhopadhyay[1,3], Nathan J. Hillson [1,2,3], Jay D. Keasling [1,3,4,6] & Hector Garcia Martin [1,2,3,7] ✉

Although synthetic biology can produce valuable chemicals in a renewable manner, its progress is still hindered by a lack of predictive capabilities. Media optimization is a critical, and often overlooked, process which is essential to obtain the titers, rates and yields needed for commercial viability. Here, we present a molecule- and host-agnostic active learning process for media optimization that is enabled by a fast and highly repeatable semi-automated pipeline. Its application yielded 60% and 70% increases in titer, and 350% increase in process yield in three different campaigns for flaviolin production in *Pseudomonas putida* KT2440. Explainable Artificial Intelligence techniques pinpointed that, surprisingly, common salt (NaCl) is the most important component influencing production. The optimal salt concentration is very high, comparable to seawater and close to the limits that *P. putida* can tolerate. The availability of fast Design-Build-Test-Learn (DBTL) cycles allowed us to show that performance improvements for active learning are rarely monotonous. This work illustrates how machine learning and automation can change the paradigm of current synthetic biology research to make it more effective and informative, and suggests a cost-effective and underexploited strategy to facilitate the high titers, rates and yields essential for commercial viability.

Although synthetic biology can produce valuable chemicals in a renewable manner, its progress is still hindered by a lack of predictive capabilities[1–3]. Synthetic biology has allowed us to produce, e.g., synthetic silk for clothing[4], meat-tasting meatless burgers using bioengineered heme[5], and antimalarial and anticancer drugs[6,7]. Furthermore, it has the potential to play a significant role in tackling climate change by enabling a circular bioeconomy[8–11], and in producing novel therapeutic drugs[12]. Its prospects are auspicious thanks to the availability of new tools for genetic editing, high-throughput phenotypic

data generation[13], and a growing demand for renewable products[14]. However, our inability to predict the outcome of engineering interventions often pushes synthetic biologists to an arduous and time consuming trial-and-error search for the optimal strain and cultivation conditions[1,15].

Media optimization is a critical step in the synthetic biology process that could significantly benefit from predictive capabilities[16,17]. Media optimization involves finding the optimal media providing the largest production levels, and is essential to obtain the high titers, rates, and

[1]Biological Systems and Engineering Division, Lawrence Berkeley National Laboratory, Berkeley, CA, 94720, USA. [2]Department of Energy Agile BioFoundry, Emeryville, CA, 94608, USA. [3]Joint BioEnergy Institute, Emeryville, CA, 94608, USA. [4]Department of Bioengineering, University of California, Berkeley, CA, 94720, USA. [5]Biomaterials and Biomanufacturing, Sandia National Laboratories, Livermore, CA, 94550, USA. [6]Department of Chemical & Biomolecular Engineering, University of California, Berkeley, CA, 94720, USA. [7]BCAM, Basque Center for Applied Mathematics, Bilbao, 48009, Spain. [8]These authors contributed equally: Apostolos Zournas, Matthew R. Incha. ✉e-mail: hgmartin@lbl.gov

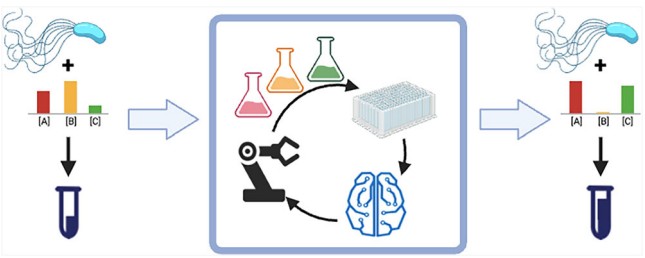

**Fig. 1 | Media optimization by combining machine learning and automation into an active learning process.** Given a strain and a media formulation (with [A], [B], [C]… concentrations for components A, B, C…) that results in a certain production of a desired metabolite, our semi-automated pipeline leverages automation, DoE, and machine learning to find the combination of media component concentrations (i.e., media design) that optimizes final production. Hence, this pipeline facilitates and automates one of the most cumbersome processes in synthetic biology. Moreover, the pipeline is generally applicable to any strain and metabolite, and can be easily translated to a cloud lab or fully automated to create a self-driving lab because of the standard hardware used and the detailed protocol provided in this paper. For this case, our media was composed of fifteen components (based on MOPS minimal medium, Table S1), out of which three components were held fixed (glucose, MOPS and Tricine), except for the yield improvement case (when glucose was unconstrained). We optimized the media to increase production of flaviolin, a precursor to a very diverse family of compounds, in *P. putida.*, an organism of versatile metabolism. Figure created using Biorender[80].

yields (TRY) that are often required for commercial viability[18]. This step is often relegated to the process optimization phase at scale-up (e.g., from 1–2 ml to 1–2 L), when all genome engineering is considered finished, because it can be very cumbersome. Indeed, the traditional and most popular approach to optimize media involves experimentally changing one component at a time, often based on biological knowledge of the host. This approach can be very time consuming for a typical media if testing all components: e.g., traditionally 10 components at 5 levels of concentration would require 50 experiments when tested one component at a time. Testing the combination of these would require $5^{10}$ experiments. It can also be ineffective if only testing a few components and our biological knowledge does not pinpoint relevant media components, and fails to consider non-linear effects (e.g., the optimal concentration of component $X$ may be different once component $Y$ is changed). Moreover, this traditional approach overlooks the opportunity for co-optimization of pathway, host, and media, which can significantly improve TRY results. Indeed, it has been shown that the optimal media for one clone can be very different than for another clone, producing differences in production as large as 70%[16,19].

Because of the limitations of the traditional approach to media optimization, more sophisticated approaches have been used, although sparsely: these include Design of Experiment (DoE) and machine learning (ML). DoE approaches[16], fitting the response to second degree polynomials using Response Surface Methodology (RSM), have been used on *P. putida* to improve phenazine-1-carboxylic acid production[20], siderophore production[21], biosurfactant production[22] as well as arginine deaminase activity[23], with varying degrees of success. RSM has also been used for media optimization in other organisms, such as to increase 2,3,5,6-Tetramethylpyrazine in *Corynebacterium glutamicum*[24]. Neural networks have been shown to outperform RSM in similar tasks[25,26]. ML has also been used to predict production of a metabolite of interest from media composition: the optimal media for the production of tyrosine and 4-aminophenylalanine in *E. coli* was found not to be optimal for growth[27], and production of rifamycin B in *Amycolatopsis mediterranei*[28] was increased by 25% testing a hundred different media compositions. These efforts show the effectiveness of machine learning models to predict performance based on media composition, yet the need for large datasets to train these models makes it prohibitive for fields where access to materials is limited or experiments are costly.

The significant cost of the large data sets needed to train ML algorithms prompts the use of active learning processes, in which ML algorithms select which data to collect. Active learning[29] uses ML in an iterative process in which the algorithm chooses the next set of experiments to be performed (i.e., the next set of instances to be "labeled"). This approach increases data efficiency dramatically, minimizing the number of experiments that need to be performed to reach the desired goal (e.g., increase production). To date, we only know of two pioneering studies that use active learning to optimize media, and neither attempted to improve synthesis of a product. Zhang et al.[30] tried to improve growth rate and growth yield in *Escherichia coli* and *Lactobacillus plantarum*, and Hashizume et al.[31] optimized cell viability for mammalian HeLa-S3 cells using the cellular abundance of NAD(P)H as a proxy. However, both of these studies required a supercomputer to handle the approximately 10 million candidate media combinations considered, making it inaccessible to most researchers with no access to large computational resources.

Flaviolin and *Pseudomonas putida* are an interesting target metabolite and host pair for production because of the metabolite's nature as precursor to a very diverse family of compounds, and the organism's versatile metabolism. Flaviolin has numerous applications in the biomanufacturing space. In general terms, flaviolin can be used as a proxy for malonyl-CoA, a precursor to polyketides and fatty acids, which have applications in the synthesis of fuels[32], materials[33], and pharmaceuticals[34]. Flaviolin is also used directly in the chemical synthesis of the napyradiomycins[35], known for their antimicrobial and anticancer activities[36,37]. Furthermore, it can be easily quantified through absorbance measurements: we have previously demonstrated its utility as a reporter for assaying the effect of glucose on Type-III polyketide biosynthesis[38]. *P. putida* displays strong tolerance to solvents and the ability to degrade aromatic compounds from lignocellulosic materials[39], a potentially very large source of carbon[40].

Here, we present a semi-automated active learning process able to optimize the culture media for production of a desired metabolite (Fig. 1), which we demonstrated on *P. putida* producing flaviolin. The approach, however, is agnostic to the molecule and host, so it is applicable to any other organisms and products. To feed the machine learning algorithm (the Automated Recommendation Tool, ART[41]) the data it needs, we created a highly repeatable, semi-automated pipeline able to test up to fifteen media combinations in just three days, requiring less than four hours hands-on time. The active learning process guided by ART produced increases of 60% and 70% in titer, and 350% in process yield in three different campaigns. Furthermore, the use of explainable Artificial Intelligence (AI) techniques pinpointed that only five of the media components strongly influenced production and, surprisingly, common salt (NaCl) was the most important. Interestingly, the optimal salt concentration for flaviolin production is very high, comparable to seawater and close to the limits that *P. putida* can tolerate. Leveraging the fast Design-Build-Test-Learn (DBTL[3,42]) cycles provided by the semi-automated pipeline, we were able to explore the behavior of active learning processes when multiple DBTL cycles are available, and showed that improvements in performance are rarely monotonous. By using synthetic data, we also showed that ART outperforms other state of the art approaches when leading the active learning process, embodying the highly-effective machine learning algorithm needed to effectively explore very large phase spaces. This work provides an illustrative example of how machine learning can be productively leveraged to accelerate and improve the biological engineering process, and suggests a cost-effective and underexploited strategy to facilitate the high TRYs essential for commercial viability.

## Results and Discussion
### Development of a semi-automated data generation pipeline provided the data quality required for machine learning to be effective
Since abundant high-quality data is critical for machine learning methods to be fully predictive[1], we invested time in developing a semi-automated pipeline able to test up to 15 media designs in triplicate/quadruplicate in a

**Fig. 2 | Our semi-automated pipeline provides the abundant high quality data that machine learning needs in order to effectively guide the engineering process. A** Media is synthesized in a BioMek NXP (Beckman Coulter) liquid handler, and is used to culture a flaviolin-producing *P. putida* strain in the Biolector Pro (Beckman) automated cultivation platform. The amount of flaviolin is assayed through its absorbance (Abs$_{340}$) in a microplate reader (Spectramax M2), and the corresponding data is uploaded to the Experiment Data Depot (EDD). ART pulls the data from EDD, trains on these data to develop a predictive model for flaviolin production (response) from the media component concentrations (input), and uses that model to recommend the media design (concentrations for all 12-13 variable components of the media) for the next cultivation. ART's media design recommendations are transformed into instructions for the BioMek liquid handler through a Jupyter notebook (See Supplementary Fig. S1 for a more detailed description). The 48 wells in the Biolector were used to test either 15 or 11 media designs per DBTL cycle. If using three replicates, 15 media designs and one control could be tested (48/3 = 16 = 15+1 control); if using four replicates, 11 media designs and one control could be tested (48/4 = 12 = 11+1 control). Each DBTL cycle takes three days to run (1 day for sample prep + 2 day cultivations), providing a convenient setup to test the effect of multiple DBTL cycles. **B** ART takes as input the concentrations of media components and as response the flaviolin titer (or yield), creates a model to predict response from inputs, and uses it to produce recommendations. Figure generated using Biorender[80].

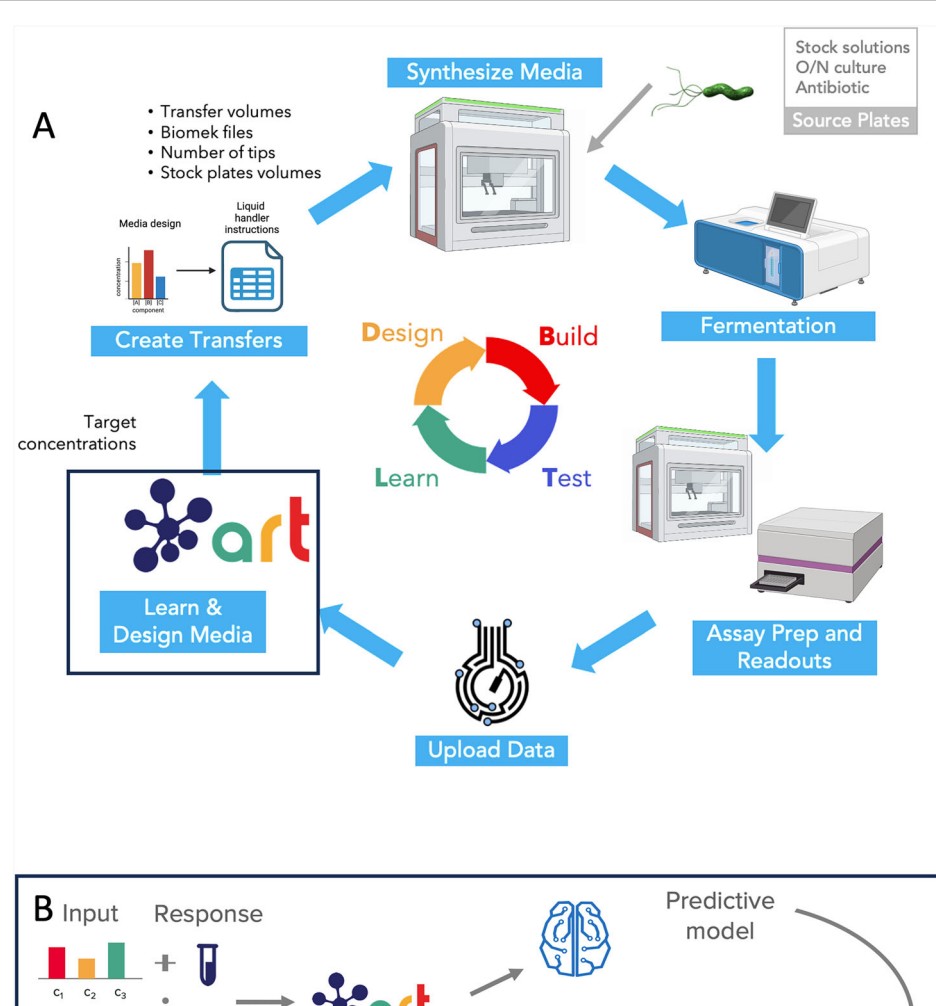

period of three days (Fig. 2, S1). Toward this goal, we automated most steps of the media optimization process. To begin with, an automated liquid handler combined stock solutions for each of the 15 media components (2–3 fixed, 12–13 variables, Fig. 1) to create media with the desired concentrations for each (i.e., a media design). Then, these media designs were dispensed in three or four wells of a 48-well plate, inoculated with the engineered *P. putida* strain, and cultivated in an automated cultivation platform. After a 48 hour cultivation, flaviolin in the culture supernatant was measured in a microplate reader using absorbance at 340 nm as a proxy. These flaviolin production data and the media designs were stored in the Experiment Data Depot (EDD)[43]. ART collected these data from EDD and used them to recommend improved media designs. These media designs, combined with the stock concentrations, were used to generate the liquid handler instructions for media preparation through a Jupyter notebook. These instructions were provided to the liquid handler along with stock solutions to build the desired media design, starting a new cycle. The Bio-Lector was chosen for cultivation because of its automated nature, the reproducibility of its data through tight control of culture conditions (O$_2$ transfer, shake speed, humidity), and its ability to produce results that scale to higher volumes[44]. In the end, media optimization in small wells is of limited use unless the results can be scaled to the higher volumes where

production will take place. The microplate reader was used because our product, flaviolin, has light absorption properties that can be measured optically, accelerating phenotype acquisition with respect to other methods (HPLC, GC-MS, etc). In this way, we used the Abs$_{340}$ as a high-throughput assay to effectively guide the active learning process, and we used the HPLC assay to validate the increases with an authoritative assay. This approach has been reported previously in Yang et al.[45], where flaviolin was used as a Malonyl-CoA biosensor and the optimal wavelength for measurement was determined to be $\lambda = 340$ nm, even though maximum absorbance is at $\lambda = {\sim}300$ nm. Final results were confirmed with HPLC (Fig. S2). To enable reproducibility through a standardized protocol description and transfer, the full protocol has been stored in protocols.io[46].

This semi-automated pipeline could be fully automated and converted into a self-driving lab[47], or transferred to a cloud lab[48,49], because it has been designed with that purpose in mind. The human labor required in this process involved the transport of strains and samples, preparation of inoculum culture, preparation of concentrated media component stock solutions, and centrifugation of cultures after production. Media preparation, inoculation, and arraying of culture supernatant into a 48-well plate were all carried out using the automated liquid handler. A fully automated platform could employ a powder handler to automate the preparation of

**Fig. 3 | The semi-automated pipeline produces replicable data.** The figures represent the 48 wells in a Biolector for DBTL cycles 1 and 2 in the first campaign. In each row (A–F) the columns 1–4 contain the same media design (technical replicates), and so do columns 5–8. The four wells in the lower right part of the plate (F 5-8) always contain the same control medium (plus a 10% added noise), to ensure that results can be replicated for different DBTL cycles. Notice how the production levels for replicates are very similar, typically with less than 10% error (Fig. S4). This repeatability is essential for machine learning methods to be effective. Any spurious deviation can produce significant deviations in predictions and recommendations.

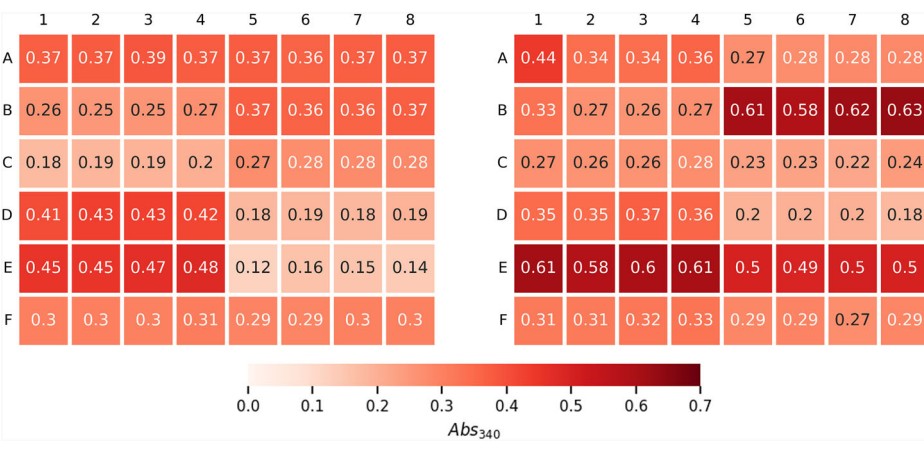

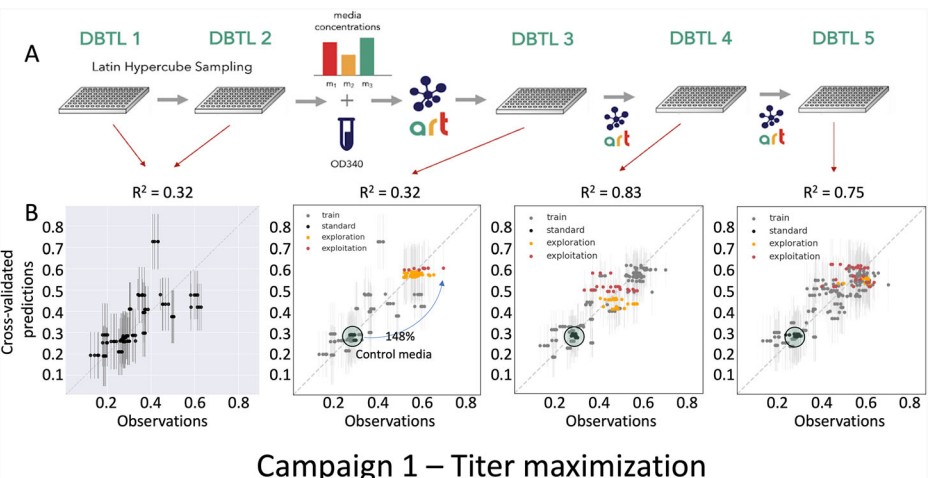

**Fig. 4 | The semi-automated active learning process generated a 148% increase in titer during campaign 1.** **A** The first 2 DBTL cycles were designed using Latin Hypercube Sampling (LHS) to explore as much of the phase space as possible. These were followed by 3 DBTL cycles, where in each cycle ART was trained on the data of the previous cycles, and generated recommendations which either maximized the variance of the output of the probabilistic model to find regions with the highest prediction uncertainty (exploration), or maximized the mean of the output of the probabilistic model to find media designs with the highest predicted flaviolin titer (exploitation). DBTL 1-2 contained each media design in four replicates. Due to high reproducibility, we decreased the number of replicates to 3 for DBTL 3-5, enabling more experiments to fit on the plate. **B** The cross-validated predictions for the first

two DBTL cycles showed limited predictive power ($R^2 = 0.32$). However, recommendations produced from this data improved the titer proxy ($Abs_{340}$) by 148% in DBTL 3. DBTL 4 and 5 did not show any improvement, but the model predictive power increased significantly in DBTL 4 with a coefficient of determination ($R^2$) reaching as high as 0.83. The fraction of exploitation (red) vs exploration (orange) recommendations increased as DBTL cycles progressed. Results for the control media (in black) for all previous DBTL cycles are included for each DBTL cycle, showing that they stay within 10% of each other (coefficient of variation = 0.067), and demonstrating repeatability between cycles. The same information for campaigns 2 and 3 can be found in Fig. S5.

stock solutions, and other automated liquid handlers could perform the centrifugation. An automated robotic arm[50] could transport strains and samples from one station to another. Complementarily, the whole process could be deployed in a cloud lab, eliminating all hands-on labor for the researcher. Furthermore, this method could be further expanded to enable quantification of metabolites requiring more sophisticated preparation, like extractions or lyophilization.

Repeatability is a key element for ML to be effective, and was tested in three ways: within a cycle (Fig. 3), between cycles (Fig. 4B) and between users (Fig. S3). For each cycle and media design, we built three to four replicates of the same media (Fig. 2). We arrived at this number after building a full plate of 48 replicates for a test media, and finding that 3-4 replicates produced an error (coefficient of variation) typically less than 10% (Fig. 3, S4). To check repeatability between cycles, we added a control to each DBTL cycle (with its corresponding replicates) representing the initial media formulation (MOPS minimal medium). We purposefully added 10% noise to the replicates of these controls' media designs to ensure the robustness of the

model, and to test the sensitivity of flaviolin production to each component concentration. As evidence of the repeatability that automation enabled, we observed that, after five to six cycles, the control medium showed no changes in flaviolin production above the 10% base noise level (black dots in Fig. 4B). Lastly, to ensure robustness of the automated pipeline to user differences (i.e. reproducibility), different users repeated the exact same protocol with the same media designs using freshly made stocks after a full year. The results were quite reproducible (Fig. S3), indicating that this process could be transferred to a different environment and maintain reproducibility.

### Applying the pipeline produced significant improvements in flaviolin production

We leveraged the semi-automated platform to perform three campaigns (C1, C2, C3, with 5-6 DBTL cycles each) and improve the titers and process yields of flaviolin production by 60% (148% as measured by $Abs_{340}$), 70% (170% in $Abs_{340}$) and 350% (300% in $Abs_{340}$) respectively (Fig. 4, Fig. S5). The first two campaigns, C1 and C2, aimed to increase the flaviolin titer

**Fig. 5 | All three active learning campaigns converged to similar regions of the media phase space.** The media phase space is a multidimensional space in which each possible media design corresponds to one unique point in the phase space. Each metabolite concentration corresponds to a coordinate axis. Due to the inherent difficulty of visualizing high dimensional spaces, we used Principal Component Analysis (PCA) to visualize the most important dimensions (along which there is maximum variation). This PCA shows that high performing media, recommended in the final DBTL cycles, lie in the same region of the phase space (red rectangle), primarily along the same value for PC1. Panels **A**, **B**, and **C** show compositions colored by DBTL cycle, while Panels **D**, **E**, and **F** show compositions colored by flaviolin production levels. Each campaign is represented by a pair of panels: (**A**, **D**): Campaign 1, titer maximization, (**B**, **E**): Campaign 2, titer maximization, and (**C**, **F**): Campaign 3, process yield maximization. The two first principal components explain 43% of the total variance. A 3D PCA is shown in Fig. S7, capturing 52% of the total variance.

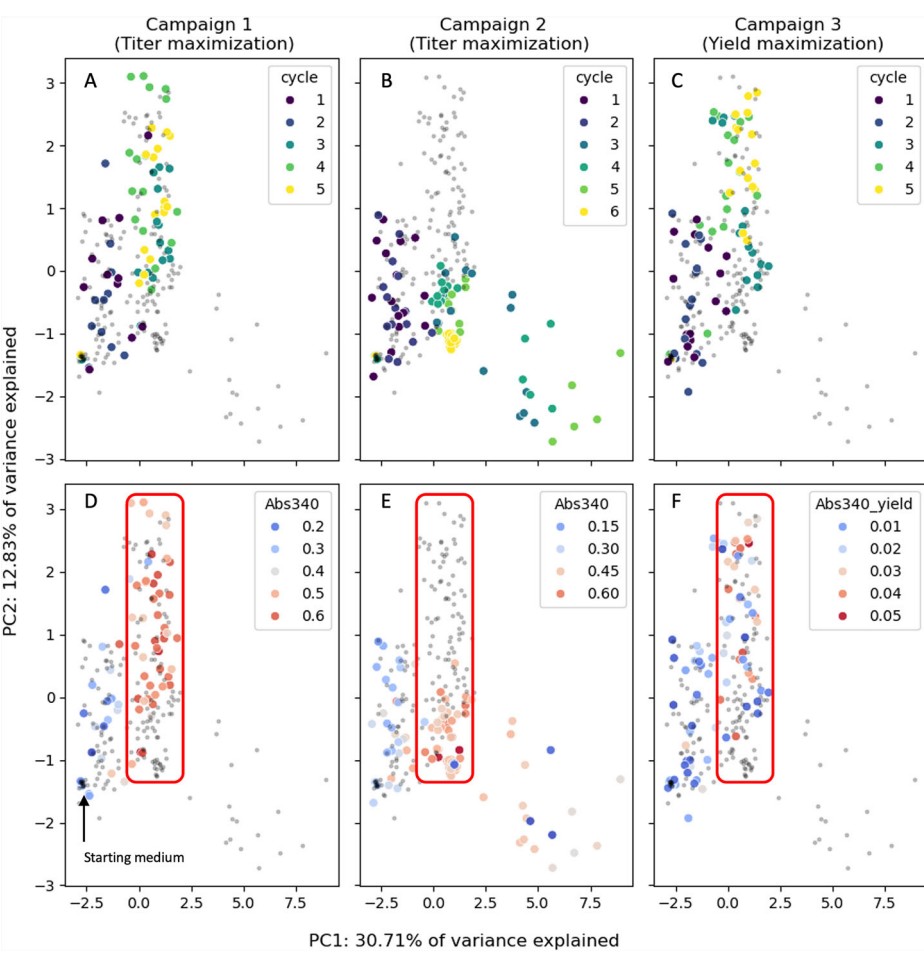

proxy ($Abs_{340}$) while keeping the glucose concentration at the same level as our baseline media. For the third campaign, C3, we aimed to increase the process yield proxy (i.e., ratio of flaviolin proxy divided by the initial glucose concentration, see Materials and Methods) after unconstraining the glucose concentration. All three campaigns converged to similar regions of the media phase space, even though the explored trajectory was different (Fig. 5). Indeed, the most successful media designs for each campaign were very similar, and displayed unexpectedly high levels of NaCl (close to the limits that *P. putida* can tolerate, Fig. S6). We evaluated the predictive accuracy of the model by using the coefficient of determination $R^2$, which represents the fraction of the response data variance explained by the model[51]. A value close to one indicates very good predictions (almost all response data explained by the model), and values close to zero or negative indicate no predictive power. Hence, a higher value of $R^2$ is desirable because it implies a higher capability to predict flaviolin production from the media component concentrations, which is critical to find the media component concentrations that maximize production.

All three campaigns were performed similarly: two DBTL cycles using DoE approaches, followed by three to four DBTL cycles of active learning guided by ART (Fig. 4, Fig. S5). DBTL cycles 1-2 were used to accumulate sufficient training data to make ART effective in predicting production from media composition. For this purpose, we used a DoE approach called Latin Hypercube Sampling (LHS), included in ART[41]. LHS does not leverage any prior biological knowledge other than the components used and their upper and lower bounds, and is a purely statistical approach producing recommendations meant to span as much phase space as possible, since ML algorithms are typically much more effective for interpolating than extrapolating. After DBTL 1 and 2, an active learning process ensued, in which we used all the previously generated data from the prior DBTL cycles to train

ART (e.g. training for DBTL 4 of C1 used the data from DBTL 1-3 of C1), and generated recommendations for the next DBTL cycle (Fig. 2). These initial 23-31 datapoints were enough to generate effective recommendations. For example, in C1 the highest performing media designs was suggested in DBTL3, from a model trained only on 23 media designs. In this active learning process we progressively shifted from an emphasis in exploration to exploitation. Explorative recommendations focused on investigating parts of the media phase space where ART's predictive power was most limited, whereas exploitative recommendations focused on suggesting new media designs that were predicted to yield the highest response[41]. We expected that ART would be able to identify high-producing media designs more accurately as a larger fraction of the media phase space was experimentally explored. As an example of this progressive shift from exploration to exploitation, in DBTL 3 from C2, 66% of the recommendations were explorative and 33% exploitative. In DBTL 4, however, 46% were explorative and 54% exploitative, and in DBTL 5, 33% were explorative and 66% exploitative (Table S2). Due to the time and monetary cost of biological experiments we limited the number of DBTL cycles performed in this study to numbers that are practical. Hence, we aimed to conduct 5 DBTL cycles per campaign. However, if we saw improvements at DBTL5, we would conduct further experiments until there was no further increase in the maximization objective for that campaign.

The first campaign (C1, Fig. 4) generated media designs that improved titer by 148%, as measured by the proxy, and 70% when measured directly (through HPLC), starting at a production of ~95 mg/l flaviolin. The active learning process in C1 started with limited predictive power at the end of DBTL cycle 2 ($R^2 = 0.32$), but was able to provide recommendations that produced in DBTL 3 the highest titer increase for the whole campaign. Predictive power increased significantly for DBTL 4 and 5 ($R^2 = 0.83$ and

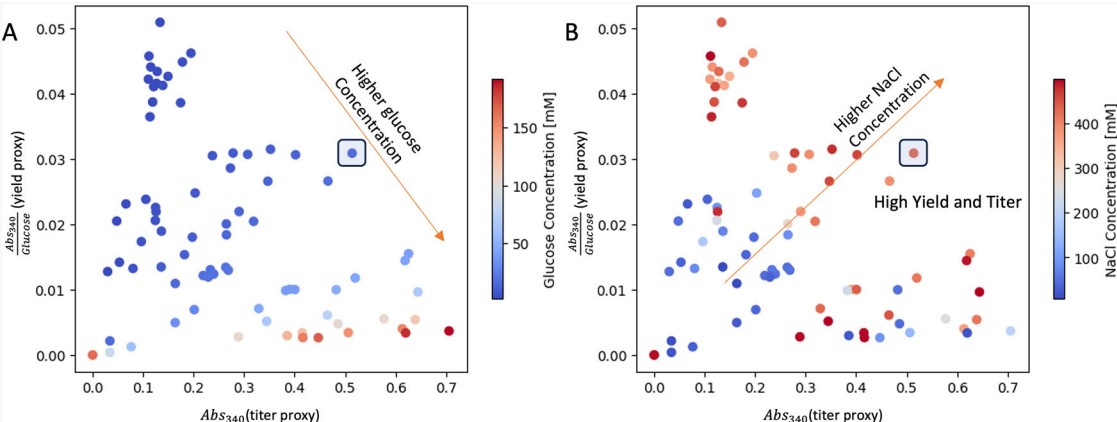

**Fig. 6 | Glucose concentration controls the tradeoff between titer and process yield, while salt determines the performance of the media designs in campaign 3.** **A** The media designs with the highest yield showed low titer, and vice-versa. **B** In either case, titer and yield were generally positively affected by higher salt concentration. Shaded square area represents a media design that shows a good compromise between titer and yield. Abs340 is a proxy for flaviolin titer.

0.75, resp.), but no further titer increases were found. DBTL cycles 1-2 of C1 used four replicates for each media design, instead of three used in DBTL 3-5 (Fig. 4), reducing the number of media designs (instances) produced relative to the initial DBTL cycles, and limiting the starting predictive power. As in the other campaigns, the repeatable titer results for all controls in all DBTL cycles (Fig. 4B) are a testament to the pipeline's excellent repeatability (Fig. 2).

Campaign two (C2, Fig. S5) produced a higher improvement than C1, increasing titer by 170%, as measured by the proxy, and 60% when measured directly, starting at a production of ~95 mg/l flaviolin. This campaign was performed to check the reproducibility of our results when a key member of the team left the institution, a common occurrence in research environments. C2 mimicked C1 except in the number of replicates used: three for all cycles instead of four for the initial two DBTL cycles in C1. The results in C1 had shown that three replicates were enough to capture the variability, and this change increased the number of media designs per DBTL cycle: from 12 (11 designs + 1 control) to 16 (15 designs + 1 control). We expected this increase in instances to improve the predictive ability of ART at each cycle. Indeed, the active learning process started with a much stronger predictive power than in C1 ($R^2 = 0.64$), but provided no increase in titer in DBTL cycles 3 and 4. It was only in DBTL 5 that the 170% increase in titer proxy was observed, along with slowly increasing predictive power ($R^2 = 0.70$). A DBTL 5 was performed with no further increase in titer, and the campaign was considered finished. However, the fact that predictive power decreased and remained around $R^2 = 0.68$ indicates that ART still did not yet have a thorough knowledge of the full phase space, and there may be areas of improved performance.

Optimizing for flaviolin process yield instead of titer generated an impressive 300% improvement of the proxy in campaign three (C3), and a 350% increase in direct flaviolin yield, starting from a yield of 0.025 (flaviolin (g) / glucose (g)) in the initial media (Fig. S5). This campaign used three replicates as well, providing good predictive power ($R^2 = 0.63$) to start the active learning process. There were no increases in DBTL 3, but DBTL 4 brought an increase of proxy yield of 300%, and the predictive power rose to the highest observed in this project, $R^2 = 0.89$. A final DBTL 5 provided no further increases and the campaign was finished, showcasing a very respectable $R^2 = 0.86$ for predicting yield. A trade-off between titer and yield is common in production strains, and a similar one between titer and process yield was found in this case (Fig. 6A): higher initial glucose concentrations improved titer and decreased yield, and vice versa. Further, a more unusual strong positive correlation of yield and titer on NaCl concentration was observed (Fig. 6B), in agreement with our observations in C1 and C2. Using these dependencies, it is possible to efficiently select yield and titer by choosing the right initial glucose and NaCl concentrations. While performing active learning to increase yield we observed a media design that

shows a good compromise between titer and yield (shaded square area in Fig. 6B). This media design was recommended by ART for DBTL 4, showing an 80% increase in titer and an 100% increase in yield from the control media.

All three campaigns converged to the same region of phase space (Fig. 5, Fig. S7), which displayed surprisingly high levels of NaCl (Fig. S6). Principal Component Analysis (PCA) showed that the active learning process in C1 and C3 followed a very similar trajectory in all cycles (Fig. 5A,C), despite different objectives. A divergence occurred in C2 after DBTL 3, however, that led to a different trajectory. Despite this difference, the best media designs were found in the same region of phase space at the end of the active learning process. The media corresponding to the highest production generally displayed concentrations of most media components 2-16 times higher than for the base media, with the NaCl concentration ten times higher (Fig. S6). This NaCl concentration (at 460 mM) is close to the limit of what *P. putida* can tolerate[52–54]. Whereas other components (e.g., $(NH_4)_6Mo_7O_{24}$) also displayed large concentration increases in the optimal media for all three campaigns, their impact in the production of flaviolin was minor, as evidenced by the feature importance analysis described in the next section.

The improvements in titer and process yield during the active learning process seem to be abrupt and unpredictable, rather than gradual. These improvements may happen when the predictive power is high (C3, DBTL 5, $R^2 = 0.63$, Fig. S5) or low (C1, DBTL 3, $R^2 = 0.32$, Fig. 4B), and they may happen with few instances (C1, DBTL 3, 23 instances, Fig. 4B) or many (C2, DBTL 5, 61 instances, Fig. S5). Hence, the lack of increased production in a given cycle does not, in any way, mean that those increases would not occur in future cycles. For this reason, it is difficult to categorically affirm that any of these titers or yields cannot be improved in the future. Indeed, the good, but not ideal, nature of our predictions at the end of the campaigns ($R^2 = 0.75, 0.68, 0.86$ for C1, C3, and C3), indicates that ART does not yet have a perfect predictive knowledge of titer or yield, and there could be remaining phase space pockets where they could be increased.

## Feature importance analysis shows that salt stress is the main driver of enhanced production

Only five out of the twelve variable components (thirteen for C3, including glucose concentration) of the media were found to play important roles in flaviolin production (Fig. 7, S8, S9): NaCl, $K_2HPO_4$, $K_2SO_4$, $FeSO_4$, and $NH_4Cl$. $K_2HPO_4$ is the sole source of phosphate in our starting medium (MOPS minimal medium), while buffering is done using MOPS and Tricine buffers. $K_2SO_4$ and $FeSO_4$ are the sulfate and iron sources, while $NH_4Cl$ is the sole nitrogen source in our media composition. NaCl provides sodium to the bacteria and also strongly regulates metabolism. Feature importance was determined through SHapley Additive exPlanations (SHAP) analysis[55],

**Fig. 7 | Feature importance for Campaign 1 shows that salts concentrations are the main drivers of production improvement.** SHAP values, indicating impact of the feature on the response, indicate that the top five of the components are the most important (SHAP values for other components are much lower). NaCl concentration in particular has the highest impact on the predicted production. High SHAP values indicate high impact in increasing response, whereas low SHAP values indicate high impact in decreasing response. Colors indicate the value of the feature for the corresponding SHAP value. Hence high values of NaCl concentration produce high values of flaviolin titer, whereas high values of $K_2SO_4$ produce low values of flaviolin titer. Similar SHAP analyses for campaigns 2 and 3 can be found in Figs. S8 and S9. SHAP values are in units of the response (Abs$_{340}$, Fig. 2).

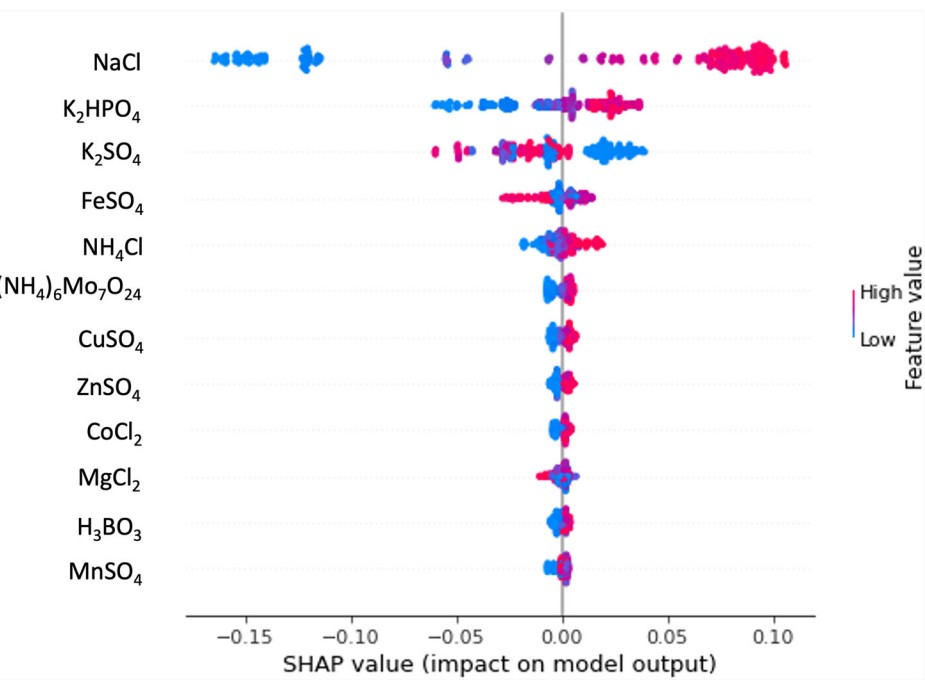

which compares the model output when a given feature is included or excluded (i.e., setting its value to the average of all observations). This process is performed for all possible features and feature combinations, with the final SHAP value being the sum of all individual feature contributions.

Surprisingly, the salt NaCl emerged as the most critical feature overall influencing flaviolin production (Fig. 7, S8, S9). NaCl ranked first in feature importance for campaigns C1 and C2, and second for C3 (with glucose being the most important, as expected). In all three campaigns, the best-performing media contained 8-9.2 times the concentration in the starting media (400-460 mM NaCl, Fig. S6). The consistency between the final results of three independent campaigns (Fig. 5), the feature importance analysis (Fig. 7), and the distinct effect of NaCl on flaviolin production (Fig. 6B) underscore the importance of NaCl for increased flaviolin production. Similar, but much smaller, increases in titer by NaCl addition have been reported in other organisms and for diverse products. For example, NaCl has been shown to improve growth as well as isoprenol production in *E. coli* in the presence of ionic liquids[56]. Increased salinity (120 mM NaCl) also improved bioinsecticide production by *Baccilus thurigiensis* combined with heat-shock[57]. Squalene accumulation in the marine protist *Thraustochytrium* sp. peaked at a NaCl concentration of 85 mM[58]. Polyhydroxyalkanoate production was boosted in NaCl concentrations of up to 154 mM in *Cuprividus necator*[59]. Lastly, in activated sludge microbial communities, high NaCl was shown to increase protease activity and decrease glucosidase activity, while reducing the microbial diversity in the process[60]. However, most of these have been ad hoc observations rather than the product of a systematic study as we do here. When a systematic approach was followed, in the case of squalene production, NaCl and glucose were found as the most important drivers of production increase[58]. However, this was a marine protist, for which the relevance of NaCl is less surprising, and only three media components were tested. Even in this case, the optimal NaCl concentration was not nearly as extreme as the one found in this study (400–460 mM NaCl) for the putative biomanufacturing host *P. putida*. Indeed, these levels are comparable with those of seawater (600 mM), and higher than the concentration used in medium for marine microalgae (308 mM NaCl), prompting the consideration of production environments as very different from growth environments.

In all three campaigns, $K_2HPO_4$ and $FeSO_4$ were in the top five most influential components for high flaviolin production. Interestingly, FeSO4 showed negative impact for most of its highest concentrations in C1, which

is the opposite trend to what we saw in C2 and C3 (Fig. 7, S8, S9). Similarly, $K_2HPO_4$ showed positive impact in the higher concentrations explored in C1, while in C2 the highest importance was observed in a "goldilocks region" within the explored concentrations, and in C3 it showed a much smaller importance. These apparent incongruencies likely resulted from the active learning algorithm exploring different parts of the phase space, changing the average explored concentration of each component, which is used to calculate a positive or negative effect (SHAP value) for each observation. In addition, given that the concentration of $K_2HPO_4$ was significantly smaller in the media providing the highest process yield, it seems likely that the phosphate demands in low glucose conditions are significantly diminished due to lower growth.

Certain components were only significant in some campaigns. For example, $K_2SO_4$ was very important in C1, $H_3BO_3$ only in C2, $NH_4Cl$ in C1 and C2, and $MgCl_2$ in C2 and C3. This variability is again due to the different trajectories the algorithm explored in the phase space. The remaining components showed minimal importance throughout all three campaigns, indicating that they are either unnecessary or required only in minimal concentrations, without adverse effects at higher levels.

Conventional wisdom based on mass-action kinetics (i.e., the need to maximize the AcCoA pool) and previous transcriptomics analyses is at odds with the result of high flaviolin production under high salinity conditions. Previous transcriptomics have shown that high salinity concentrations significantly affect the central carbon metabolism in *Pseudomonas* species. In these transcriptomics studies the following was observed: flaggela-related proteins were down-regulated, indicating a tendency to generate a biofilm in order to respond to osmotic stress; N-acetylglutamylglutamine amide (NAGGN) biosynthesis was upregulated, as this metabolite is one of the most prominent osmoprotectants; membrane composition was changed by overexpression of cardiolipin; and the expression of siderophores was upregulated as iron-carrying proteins are used to combat Reactive Oxygen Species (ROS) stress (a common side-effect of osmotic stress)[61]. Similar results were found in *P. aeruginosa*, where NAGGN biosynthesis knockouts lost their ability to grow in NaCl concentrations of 500 mM, but this phenotype was rescued by adding betaine in the growth media[62]. However, cardiolipin requires glutamine and G3P to be synthesized, and NAGGN requires AcCoA and Glutamine. Glutamine consumption would pull carbon away from the production of flaviolin, which requires malonyl-CoA, and similar effects would be expected by pulling carbon from G3P and

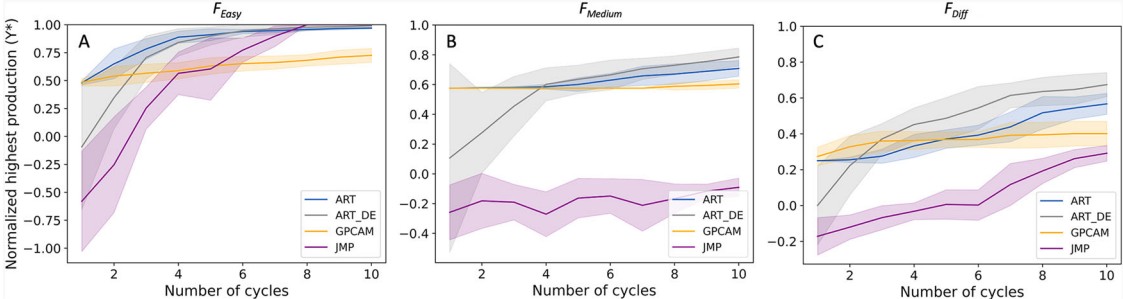

**Fig. 8 | ART outperforms other state-of-the-art algorithms (JMP and gpCAM).** We used ART, JMP, gpCAM to guide three corresponding simulated active learning processes where the 15 input variables (media designs) and responses (flaviolin production) were used to recommend the next set of media designs. Response was simulated through three different functions that present different levels of difficulty to being "learnt". These functions of increasing difficulty are: **A** $F_{Easy} = \frac{1}{d}\sum_i^d (x_i - 5)^2 + e^{-\sum_i^d x_i^2} + 25$, **B** $F_{Medium} = \frac{1}{d}\sum_i^d x_i^4 - 16x_i^2 + 5x_i$ and **C** $F_{Diff} = \sum_i^d \sqrt{x_i}\sin(x_i)$ The starting media designs were the same for all algorithms and each process was run for

10 DBTL cycles. Each process was run ten different times, and the lines and shaded areas represent the mean and standard deviation of the highest production for each cycle. Individual traces for each run are shown in Fig. S10. Y* represents the maximum production at each cycle normalized by the true optimum of each function. We also tested a new recommendation algorithm for ART that improves on its original parallel tempering approach (ART_DE). ART and ART_DE reached the highest productions after 10 DBTL cycles, except in the case of $F_{Easy}$, where the difference with the best algorithm (JMP) is minimal.

AcCoA. Hence, one would expect the production of flaviolin to decrease, rather than increase under high-salinity conditions, because carbon is being pulled away from the production of the precursors to flaviolin. This highlights how utilizing a purely data-driven approach can produce results and insights that are not obtainable through standard metabolic engineering approaches.

## ART outperforms other state-of-the-art approaches for guiding the optimization process

Because the challengingly wide phase space targeted in this project demanded the most efficient algorithm, we used synthetic data to compare ART's performance with two state-of-the-art alternatives: JMP[63] and gpCAM[64] (Fig. 8). JMP implements the RSM, which relies on a second degree polynomial, to design and augment experimental datasets, and it has been the default choice for process optimization through DoE. gpCAM implements gaussian processes to model the data and a global optimization algorithm to generate recommendations, and it has been used for automated experimentation in a synchrotron facility[65]. We also compared a new approach for generating ART recommendations: differential evolution, which is an evolutionary algorithm used for global optimization, and a part of the scipy.optimize suite[66,67]. We used synthetic data for this comparison because full experimental tests are expensive and time consuming, and these comparisons give a good estimate of the relative efficiency of methods. For each algorithm we simulated 10 DBTL cycles mimicking the experimental process used in this study: we started with 16 instances with 15 input variables, and known response values. For each cycle, each of the algorithms was trained with the data collected so far and was used to provide 16 recommendations. These recommendations were used to generate responses using three functions of different difficulty to learn: an easy, a medium and a difficult one (Fig. 8), similarly to our previous work (Fig. 4 in Radivojevic et al.[41]). For each algorithm and difficulty level, the process was performed 10 times and averaged, so as to eliminate the effects of stochasticity.

ART performs overall better than both JMP and gpCAM when guiding an active learning process simulated through three functions of varying complexity (easy, medium, and difficult) (Fig. 8). For the easy function, JMP outperformed both ART and gpCAM, and is the only algorithm to predict the exact variable values for optimal production (with ART a very close second). This is probably due to the fact that the RSM used in JMP assumes quadratic interactions between terms, which is a very good approximation of the easy function. gpCAM performed the worst of the three algorithms for the easy function. In both the medium and difficult functions, ART outperformed both gpCAM and JMP. For the medium case, JMP struggled to even reach positive responses, and gpCAM improved response only linearly

and very slowly. For the difficult case, JMP's performance improved, but was not able to reach better responses than gpCAM or ART. As previously, in this case gpCAM only saw a very slow increase in response as more DBTL cycles accrued. Since JMP uses a quadratic approximation, when functions deviate from that form its performance is limited. On the other hand, gaussian processes are able to handle functions of arbitrary form, allowing for higher versatility and hence outperforming JMP in the medium and difficult cases. Lastly, ART uses an ensemble model, which includes a gaussian process among other algorithms. Since the ensemble performs as well or better than any of its constituting models, ART performs better than gpCAM. We also tested a new approach to select recommendations through ART that uses an evolutionary algorithm[67], instead of parallel tempering[41], to recommend the next set of media designs. The improved results from the evolutionary algorithm approach relative to parallel tempering warrant the inclusion of this method as part of the ART package from now on.

In summary, ART required fewer data points for similar outcomes, or achieved better outcomes altogether, enabling the ambitious active learning approach taken here. These advantages are very important for applications in which data is expensive to acquire (such as synthetic biology). The availability of a very effective predictive algorithm enables the approach taken in this project, in which we explored a very large phase space with 12-13 variable components spanning 1-2 orders of magnitude each.

## Conclusion

We have shown that active learning can optimize media in a systematic way that is agnostic to host, product, and pathway (Fig. 1), and can both generate surprising increases in production and identify unexpected key media components. To enable this active learning approach, high quality data are required, which we generated by creating a semi-automated pipeline (Fig. 2). This semi-automated pipeline enabled us to test up to 15 new media designs in triplicates within 3 days of experimentation (including a 48 h cultivation) with high repeatability within the same DBTL cycle (Fig. 3), between DBTL cycles (Fig. 4B) and between users and preparations (Fig. S3). Applying active learning to this pipeline produced 148% and 170% increases in titer and a 300% increase in process yield in three different campaigns (Fig. 4). When measured directly, without relying on the proxy used to guide the active learning process, titer increases were 70% and 60%, respectively, and the yield increase 350%. The active learning algorithm converged to similar regions of the phase space in all three campaigns even though the explored trajectories were different (Fig. 5), finding that media composition for maximal production differed significantly from the one traditionally used for assaying growth phenotypes. When we allowed the glucose concentration to change and optimized for process yield, we found that there is a tradeoff between titer and yield. This tradeoff was less

prominent when using high NaCl concentrations (Fig. 6). Only five out of the twelve (or thirteen) media components strongly influenced production and, unexpectedly, NaCl was the most important (Fig. 7). The concentration of NaCl had been previously known to affect production levels in other cases, but not shown to be the most important driver, and optimal values were not nearly as extreme as the one found in this study. We also showed, using synthetic data, that ART outperforms a widely used DoE approach (JMP) and other machine learning approaches (gpCAM) when leading the active learning process using synthetic data. ART requires fewer data points for similar outcomes or achieving better outcomes altogether (Fig. 8). These advantages are critical in cases where the cost of and time for data acquisition is high, such as in synthetic biology. The novelty of this approach resides not so much in the amount of data being produced, but rather in how the data being produced is very efficiently leveraged by ART to guide an active learning process that effectively improves production by optimizing media, a problem that every bioengineer faces.

The differences between $Abs_{340}$ and analytical quantification through HPLC indicate that, contrary to what has been previously reported[45], $Abs_{340}$ is not a very accurate proxy for flaviolin, especially when the cells are grown in significantly different physiological conditions. However, the linear correlation between the $Abs_{340}$ proxy and the HPLC (Fig. S2) shows that the use of $Abs_{340}$ can help identify better media designs. While using HPLC for all measurements would have resulted in more accurate measurements, it would have also slowed down the experimental work and it is not clear that we would have been able to identify media compositions that enabled almost double flaviolin production. Hence, a proxy, even if imperfect, can be more desirable than a more exact measurement if it accelerates the active learning process and the ML algorithm can manage the noise.

The ability to leverage fast DBTL cycles through automation allowed us to explore how active learning processes behave when several cycles are available. In our three campaigns, we have seen that the response does not improve in a monotonous fashion. Rather, improvements in the response happened in bursts and in a rather unpredictable fashion (Fig. S10): they may happen when we have collected many instances or a few, and when our predictive power is either high or low. This behavior is in agreement with other active learning studies published recently[68,69]. Hence, it is not trivial to decide when to finish a campaign. Due to the inherent stochasticity of this process, similar active learning processes (e.g. C1 and C2) will not follow the same trajectory in phase space, even if they end up converging on the same region. Experimental repeatability between biological replicates and DBTL cycles is critical for active learning to work properly: different responses for the same input can seriously diminish the predictive power of the algorithm and compromise the quality of the recommendations for the next cycle. When performed manually by a researcher, this active learning process is both time consuming and error prone: the preparation of 48 wells including 15 media designs with 12 variable components used here requires approximately 800 liquid transfers of varying volumes. Our carefully designed semi-automated pipeline (Fig. 2) provided the repeatability needed inside each cycle (Fig. 3), between cycles (Fig. 4B), and between users and preparations (Fig. S3). Moreover, it minimized hands-on time (approximately 1 hour for sample prep, 30 minutes for measuring $OD_{600}$ and $Abs_{340}$ and 2 hours for data analysis). In the midst of the repeatability crisis[70], such repeatability and reproducibility tests are crucial for biological research, especially in the context of machine learning. Furthermore, the low experimental noise provided by automation allowed for a smaller number of replicates, increasing the number of instances available for training and, eventually, increasing the predictive power. Finally, active learning can produce quite unexpected results if the phase space is made as wide as possible. However, a powerful predictive algorithm is required to efficiently search through this large phase space. In this case, the high predictive power and versatility of ART (Fig. 8), provided by the ensemble model design, and the quality of its recommendations, allowed us to explore a very large phase space with 12 (13 in campaign 3) variable components spanning 1-2 orders of magnitude each, even when starting from a very small training dataset (e.g. at DBTL3 of C1). This approach not only allowed us to optimize media

while minimizing the number of (relatively expensive) experiments, but also to identify media designs that would not have been accessible using less powerful approaches, which might have required fully constraining some media components not expected to be important (e.g., NaCl). In this study, optical measurements ($Abs_{340}$) correlated well with analytical measurements (HPLC), showing a coefficient of determination ($R^2$) equal to 0.74 (Fig. S2). Even though this correlation is not perfect, performing optical measurements allowed us to perform fast DBTL cycles, enabling rapid improvements in titer and yield, which were confirmed through the HPLC measurements (the golden standard). More accurate measurements might enable higher or faster improvements. Solutions like the Agilent RapidFire-MS or (ultra)-high-performance liquid chromatography (UHPLC) could bridge the gap between fast and accurate and fast measurements that enable machine learning in biology.

The use of machine learning to optimize media generated interesting insights about *P. putida* metabolism. The optimal media for flaviolin production was very different from traditionally used growth media. Surprisingly, high salinity was consistently the most important factor increasing flaviolin production, with the optimal media designs containing concentrations of NaCl higher than 400 mM. The other two consistently important factors, $K_2HPO_4$ and $FeSO_4$, were the iron and phosphate sources in the media. All these media components represent inexpensive additions to media that improve flaviolin production. The optimal salt concentration is comparable to seawater, opening up the possibility of considering brackish water for biomanufacturing using *P. putida* as a host, significantly decreasing the cost and environmental impact of biomanufacturing processes. Interestingly, we did not see a significant growth penalty with an increase in $Abs_{340}$ (Fig. S11), showing that *P. putida* grows reasonably well in high NaCl concentrations. The study of flaviolin itself offers interesting clues to *P. putida* metabolism because it is a proxy for its precursor, malonyl-CoA. Malonyl-CoA is central to metabolism and enables the biosynthesis of several industrially significant bioproducts, such as fatty acids. Indeed, the optimal salt concentration recommended by the machine learning approach runs counter to what traditional reasoning based on mass-action kinetics and available transcriptomics data would suggest to increase flux through malonyl-CoA to flaviolin. The main osmoprotectants generated require G3P, AcCoA, and glutamate and glutamine to pull carbon away from central metabolism, limiting the supply available for malonyl-CoA synthesis.

This work provides an illustrative example of how machine learning can be used to accelerate and improve the biological engineering process. The traditional approach for media optimization would require producing a hypothesis from empirical biological knowledge or the literature (e.g., phosphate is an important part of biomass, which can impact production) and then performing experiments to test the hypothesis (e.g., measure production under five different levels of $K_2HPO_4$). The ML-guided approach used here is different: it aims to check the impact of media on production by generating a high-throughput pipeline that explores a wide phase space of all possible chemical concentrations (even for components that we do not expect will make a large difference). The generated data is then used to guide an active learning process that pinpoints which inputs are most important and, ultimately, the best media design. Both approaches use the scientific method of hypothesis generation and experimental testing, but the hypothesis used in the ML-guided approach is more general and wide, and ultimately produces better outcomes, and even unexpected results. The use of this approach for synthetic biology in general (e.g., pathway optimization or host engineering through gene interference) can significantly accelerate its timelines. This work also goes beyond previous applications of active learning to media optimization[31] by showing that this approach can significantly improve production of valuable chemicals, that the most important production drivers can be inexpensive and unexpected components, that automation can significantly improve the active learning process, and that efficient recommender and prediction algorithms can make this active learning process run on a laptop instead of a supercomputer.

Several possibilities for future work are suggested by this study. This media optimization process is fast and effective enough that it can be

**Table 1 | Strains and plasmids used in this work**

| Strains & Plasmids | Genotype | Source | JBEI Part ID |
|---|---|---|---|
| **Strains** | | | |
| *E. coli* XL1-Blue | K12 *recA1 endA1 gyrA96 thi-1 hsdR17 supE44 relA1 lac* [F⁻ *proAB lacIqZΔM15* Tn*10* (*tetR*)] | Agilent | |
| *E. coli* S17-1 | K12 F- *RP4-2(Km::Tn7,Tc::Mu-1) λpir⁺ recA1 endA1 thiE1 hsdR17 creC510* | ATCC 47055 | |
| *P. putida* KT2440 | wildtype | ATCC 47054 | |
| *P. putida* sART1 | *P. putida* KT2440 PP_5404-06::PhiC31$_{attB}$ | This work | JPUB_025894 |
| *P. putida* sART2 | *P. putida* KT2440 PP_5404-06::PhiC31$_{attB}$::pIS100 | This work | JPUB_025892 |
| **Plasmids** | | | |
| pBADT-*rppA*-NT | pBADT-*kanR*-P$_{araBAD}$-*rppA*(NT) | Incha et al.[38] | JPUB_016949 |
| pMQ30 | *gentR sacB* | Shanks et al.[79] | |
| pMQ30-attB | pMQ30-*gentR*-PP_5404-06::PhiC31$_{attB}$ | This work | JPUB_025898 |
| pIS1 | pColE1-*ampR*-P$_{tac}$-*phiC31* | This work | JPUB_025896 |
| pIS100 | pColE1-*kanR*-PhiC31$_{attP}$-P$_{J23100}$-*rppA* | This work | JPUB_025895 |

performed simultaneously with pathway or host optimization, leading to different optimal media for different production strains. Also, it would be desirable to use AI approaches to expand the phase space: i.e. suggest new components for the media to increase production, based on extant literature. In addition, due to the process being easy to learn, repeatable, and low cost, it can be either used as a teaching template for machine learning and automation, or as a testbed for self-driving labs (SDLs).

In sum, this work illustrates how ML and automation can change the paradigm of current synthetic biology research to make it more effective and informative, and provides a cost-effective and underexploited strategy to facilitate the high TRYs essential for commercial success.

## Materials and Methods
### Chemicals, media, and culture conditions
Standard inoculum cultures of *Pseudomonas putida* KT2440 and *Escherichia coli* strains were conducted in LB medium at 30 °C and 37 °C respectively with the addition of appropriate antibiotics. Antibiotics employed were carbenicillin (100 mg/L), kanamycin (50 mg/L), gentamicin (30 mg/L), and chloramphenicol (25 mg/L) all sourced in concentrated solution from Teknova. MOPS minimal media components: MOPS, tricine, FeSO$_4$•7H$_2$O, NH$_4$Cl, K$_2$SO$_4$, MgCl$_2$, NaCl, (NH$_4$)$_6$Mo$_7$O$_{24}$•4H$_2$O, H$_3$BO$_3$, CoCl$_2$, CuSO$_4$, MnSO$_4$, ZnSO$_4$ were all sourced from Millipore Sigma. MOPS and tricine solutions were brought to pH 7.5 by the addition of 5 M or solid KOH. The flaviolin standard was prepared via preparative thin-layer chromatography (TLC) from *P. putida* KT2440 pBADT-*rppA*-NT extract as described previously[38].

### Strain and plasmid construction
Plasmids and primers were designed using the Device Editor and Vector Editor implementation of J5[71,72]. PCR products were amplified using Q5 polymerase following the supplier's instructions (NEB). All plasmids were constructed using NEB-HiFi reaction mix following the supplier's instructions using either purified PCR product or 0.5 µL of each PCR reaction directly following DpnI digest (NEB). NEB-HiFi assembly mixtures were then transformed via heat shock into *E. coli* XL1-Blue chemical competent cells (Agilent). Plasmids were purified from *E. coli* via Qiaprep Spin Miniprep kit (Qiagen) and sequenced (Azenta Life Sciences). *E. coli* S17 was transformed via electroporation of purified plasmid DNA[73].

Construction of the PP_5404-5406::PhiC31$_{attB}$ strain (sART1) was conducted in the same manner as has been described previously for genetic knockouts in *P. putida* KT2440 except with the PhiC31$_{attB}$ site between the homology arms[74]. Briefly, *P. putida* KT2440 was conjugated with *E. coli* S17 carrying the knockout plasmid (pMQ30-*gentR*-PP_5404-06::PhiC31$_{attB}$). SacB/sucrose counterselection then was used to select for removal of the plasmid backbone and integrants were screened via PCR and the PCR products sequenced (Azenta Life Sciences). For integration of a constitutive

rppA expression cassette, we constructed pColE1-*ampR*-P$_{tac}$-*phiC31* (pIS1) and pColE1-*kanR*-PhiC31$_{attP}$-P$_{J23100}$-*rppA* (pIS100). Plasmids pIS100 and pIS1 were mixed (100 ng each) and co-transformed via electroporation into sART1 yielding the constitutive flaviolin producing strain, sART2[75]. Again, transformants were screened via PCR and PCR products sequenced (Azenta Life Sciences).

All strains and plasmids are listed in Table 1 and are available through the JBEI public registry (https://public-registry.jbei.org/folders/874).

### Development and use of an automated platform for media preparation
In order to enhance reproducibility, the full protocol can be found in protocols.io[46]: https://doi.org/10.17504/protocols.io.81wgbx7eylpk/v2 includes the protocol to run the experiment and https://doi.org/10.17504/protocols.io.x54v9pr51g3e/v2 includes the protocol for measurement of Abs$_{340}$ and OD$_{600}$.

The first step in using the active learning pipeline (Fig. 2, S1) involves deciding on the strain to be used, the media components and, importantly, the lower and upper bounds for each component. In this case, we selected a flaviolin-producing *P. putida* strain (see above) and the widely used, well-defined MOPS minimal media, in which we varied 12 components in C1 and C2 and 13 components in C3 (Table S1). In preparation for the media optimization processes, we used the media compiler library (https://github.com/JBEI/media_compiler) to calculate the stock concentrations to be mixed to build the media designs. This library takes into account the solubility and concentration bounds for each component, as well as physical limitations of the equipment (e.g. minimum transfer volume), and calculates two stock concentrations (a high and low concentration) for each component. Following this, based on each recommended media design, the pipeline outputs the required volume of each stock concentration for each component, as well as the required files to perform the transfers using the Biomek NX-S8.

In the initial **Design** phase of the DBTL cycle, we generated 22 initial media designs in C1 and 30 initial designs in C2 and C3 generated using Latin Hypercube Sampling, and split these in 2 DBTL cycles, DBTL 1 and DBTL 2 (Fig. 4, Fig. S5). The media designs, along with the stock concentrations, were used to calculate the required volume from each stock using the media compiler, and generate the liquid handler instructions. The Create_Transfers.ipynb notebook in the media compiler library creates the liquid handler instructions as follows: first, the amounts of each stock solution required for each media design are computed. Then, the source and destination wells are provided in a .csv file along with the volumes for each transfer. Different liquid handlers will require different formatting, the current output is designed for use in a BioMek NXS8. The controls in each DBTL cycle were based on the MOPS minimal media with random changes (up to ±10%, uniformly distributed) in the concentration of each of the variable components (12 for C1 and C2 and 13 for C3).

In the **Build** step of the DBTL cycle, the stock plates were loaded onto the liquid handler (Biomek NX-S8) along with the culture inoculum and an empty Biolector 48-well flower plate, and 1.5 mL of media + inoculum was dispensed directly in the Biolector plate. The inoculum was prepared from an overnight culture, which was grown at 30 °C. Kanamycin (50 µg/mL) was used both for the overnight culture and the Biolector cultivation.

In the **Test** step of the DBTL cycle, the strain cultivation took place in the Biolector Pro (Beckman) for 48 h at 30 °C, 800 rpm, and 80% relative humidity according to the manufacturer's instruction for optimal oxygen transfer. Immediately post-cultivation, the liquid handler was used to prepare plates for absorbance measurements. For supernatant analysis, the liquid handler aliquoted 1 mL of culture into a 96 deep-well plate and spun at 3200 x g in a centrifuge (Eppendorf 5810 R). 200 µL of remaining culture and 200 µL of culture supernatant from the deepwell plate were aliquoted into black-sided, clear-bottom 96 well plates. $Abs_{340}$ and $OD_{600}$ were measured as a proxy for flaviolin and cell density, respectively, using a Spectramax M2 microplate reader (Fig. 2). In campaign 3, the yield used as optimization target was the effective process yield (or process yield): i.e., the ratio of flaviolin divided by the initial glucose concentration. We used effective process yield because it is more amenable to high-throughput assays, and we believe it is most relevant to biomanufacturing purposes: it is the process yield that is typically used in technoeconomic analyses. The data, along with the media design information was then uploaded to the EDD[43]. In each cycle, we identified two types of outliers, likely stemming from errors by the liquid handler: 1) cases where one of three replicates did not produce any biomass or flaviolin, possibly due to an error when inoculating the well, and 2) cases where one triplicate had significantly higher $Abs_{340}$ than the others, possibly due to transferring cells along with the supernatant. In both of these cases these media designs were removed from the pipeline and were not used to train ART. Creating the EDD study and data files is described in the notebooks "DBTLX_Create_EDD_Study_Files.ipynb".

In the **Learn** step of the DBTL cycle, the data from DBTL 1 and DBTL 2 were retrieved from the EDD and fed into ART to train its probabilistic model[41]. In the **Design** step for the next cycle, the phase space was sampled using parallel tempering[41,76] (C1 and C2 DBTL 3) or differential evolution[66,67] (C2 DBLT4-6 and C3), to recommend media designs that maximize either flaviolin production (exploitation) or model uncertainty (exploration). The number of exploration and exploitation recommendations generated by ART in each DBTL cycle is shown in Table S2. The notebooks "DBTLX_C_ART_Media_Designs.ipynb" describes training ART and generating recommendations. Generating the instructions for the liquid handler is shown in the notebooks "DBTLX_D_Create_Transfers.ipynb".

In every DBTL cycle after DBTL 2, the model was trained on all the data generated from earlier DBTL cycles in its respective campaign (e.g. in DBTL 4 of C2, the model was trained using the data generated in DBTL 1 - DBTL 3 of C2). We then proceed to the **Build** step and iterate for the next DBTL cycles (5 DBTL cycles for C1 and C3 and 6 DBTL cycles for C2).

## HPLC method for flaviolin quantification

Supernatants from 48-well flower-plate cultivation were diluted in an equal volume of methanol with 15 mg/L bisdemethoxycurcumin as an internal standard. Analysis was conducted on an HPLC (Agilent 1200 series) with a diode array detector (Agilent Technologies, USA). A Kinetex C-18 column was used for separating the analytes with no temperature control (approx. 20 °C) (2.6 µm diameter, 100 Å particle size, dimensions 100 × 3.00 mm, Phenomenex, USA). Water + 0.1% formic acid (A) and methanol + 0.1% formic acid (B) were used as the mobile phase. 5 µL injection volume and a constant flow of 0.4 mL/min were used. The following gradient was used: 0-1 min 70% A, 1–10 min 70–30% A, 10–20 min 30–17.5% A, 20–21 min 17.5–70% A, 21–26 min 70% A. Flaviolin was measured at 300 and 320 nm, and bisdemethoxycurcumin internal standard was measured at 440 nm. Flaviolin eluted at approximately 7.9 min, and bisdemethoxycurcumin eluted at approximately 13.7 min. Discrepancies in the increases in flaviolin concentration compared to the plate reader absorbance measurements are

due to the linear fit of the comparison of these data sources having a nonzero y intercept (Fig. S2). This is likely the result of differences in the limits of detection between the two measurements. Indeed, the good but not too high correlation between $Abs_{340}$ in our culture supernatant and HPLC measurements ($R^2 = 0.74$), indicate that $Abs_{340}$ is, at best, a semi-quantitative proxy for flaviolin. This is especially true when cells are grown in very diverse physiological conditions, since there is background $Abs_{340}$, (Fig. S1[45])

## Feature importance to elucidate factors with large effects on bioproduction using Shapley analysis

Feature importance was calculated using the SHAP python package[55]. SHAP uses the trained model to calculate the most important components of the media for flaviolin production. The model used for this analysis was trained on all 6 DBTL cycles, predicting flaviolin production based on the concentration of the media components. SHAP values were calculated based on the trained model and the training dataset. The *Explainer* class of the SHAP package was used as this was the most appropriate mode for an ensemble model. SHAP analysis is shown in the "DBTL1-5_Analysis.ipynb"/ "DBTL1-6_Analysis.ipynb" found in the repository of each campaign.

## Comparison with other algorithms

For each algorithm, we performed 10 learning cycles, starting from 16 observations with known function values, replicating the physical pipeline used in the rest of this work (Fig. 2). In each cycle we produced 16 new recommendations, used them as input in the benchmarking functions (Fig. 8), and used the responses obtained as the observations for the next cycle. To address the intrinsic stochasticity of the algorithms, we performed this process 10 times starting from the same initial observations, and calculated the mean and standard deviation of the highest suggested performance in each cycle.

When benchmarking ART and gpCAM, we used an exploration-exploitation tradeoff when generating recommendations. The objective function maximized to select the recommendations was:

$$G(x) = (1 - \alpha)E(y) + \alpha Var(y) \qquad (1)$$

Where x is the vector of input variables, y = y(x) is the response variable, and E(y) and Var(y) denote the expected value and variance of the probabilistic model output[41]. The exploration-exploitation trade-off parameter $\alpha = 0.1*N$, with N being the number of the DBTL cycle. Hence, in the first cycle, the objective function was

$$G(x) = Var(y) \qquad (2)$$

While in the 10th cycle the objective function was

$$G(x) = E(y) \qquad (3)$$

JMP does not support this tradeoff, and the maximum predicted production is reported for each cycle.

ART is based on an ensemble method, which combines multiple models, out of which 2 of them were TPOT models[77]. gpCAM was trained using the default gaussian kernel and JMP was trained using the Response Surface Methodology, which is a quadratic approximation of the response surface. gpCAM training is shown in the notebook "gpcam_benchmark_-funcs.ipynb", and ART using parallel tempering training was performed using the "ART_benchmark_runs.py". Differential evolution training is shown in the notebook "ART_DE_benchmark_runs.ipynb" included in the repository for C1.

## Reporting summary

Further information on research design is available in the Nature Portfolio Reporting Summary linked to this article.

## Data availability

The experimental data can be found in the corresponding repository for each study named "DBTL1-5.csv" for C1 and C3 or "DBTL1-6.csv" for C2. The data can also be accessed through the JBEI instance of the Experiment Data Depot[43] (public-edd.jbei.org), in the following studies:

Campaign, DBTL cycle, Link
Campaign 1, DBTL1, https://public-edd.jbei.org/s/flav_c1_dbtl1
DBTL2, https://public-edd.jbei.org/s/flav_c1_dbtl2
DBTL3, https://public-edd.jbei.org/s/flav_c1_dbtl3
DBTL4, https://public-edd.jbei.org/s/flav_c1_dbtl4
DBTL5, https://public-edd.jbei.org/s/flav_c1_dbtl5
Campaign 2, DBTL1, https://public-edd.jbei.org/s/flav_c2_dbtl1
DBTL2, https://public-edd.jbei.org/s/flav_c2_dbtl2
DBTL3, https://public-edd.jbei.org/s/flav_c2_dbtl3
DBTL4, https://public-edd.jbei.org/s/flav_c2_dbtl4
DBTL5, https://public-edd.jbei.org/s/flav_c2_dbtl5
DBTL6, https://public-edd.jbei.org/s/flav_c2_dbtl6
Campaign 3, DBTL1, https://public-edd.jbei.org/s/flav_c3_dbtl1
DBTL2, https://public-edd.jbei.org/s/flav_c3_dbtl2
DBTL3, https://public-edd.jbei.org/s/flav_c3_dbtl3
DBTL4vhttps://public-edd.jbei.org/s/flav_c3_dbtl4
DBTL5, https://public-edd.jbei.org/s/flav_c3_dbtl5

Freely available accounts on public-edd.jbei.org are required to view and download these studies.

## Code availability

The media compiler and notebooks included require Python 3.10 and the following packages: scipy 1.13.0, numpy 1.26.4, pandas 2.2.2. Jupyter notebooks are available at https://github.com/JBEI/Flaviolin_media_opt_C1, https://github.com/JBEI/Flaviolin_media_opt_C2, https://github.com/JBEI/Flaviolin_media_opt_C3 (for campaigns C1, C2 and C3) and https://github.com/JBEI/media_compiler for the media compiler, under the LBNL open source license. In order to run the notebooks of each campaign, a functional ART installation is required. ART is free for academic use and a license can be requested as explained in https://github.com/JBEI/ART#license. Static versions of these repositories can be found in the following Zenodo[78] links:

Media compiler: https://doi.org/10.5281/zenodo.15093709
Campaign 1: https://doi.org/10.5281/zenodo.15093357
Campaign 2: https://doi.org/10.5281/zenodo.15093361
Campaign 3: https://doi.org/10.5281/zenodo.15093363
For each campaign, the following notebooks are included:
In order to calculate the required stock concentrations and create the stock plate templates for the liquid handler, the first two notebooks need to be run before the first DBTL cycle:
"DBTL0_A_Find_Stock_Concentrations.ipynb"
"DBTL0_B_Create_Stock_Plates.ipynb"
For the first two DBTL cycles the initial recommendations were generated using the notebook "DBTL1_2_Initial_Media_Designs.ipynb" while for the next DBTL cycles the recommendations were generated using the notebooks "DBTLX_C_ART_Media_designs.ipynb".
Liquid handler instructions, files to upload to EDD and analysis was performed for each cycle using the notebooks:
"DBTLX_D_Create_Transfers.ipynb"
"DBTLX_E_Create_EDD_Study_Files.ipynb"
"DBTLX_F_Analysis.ipynb"

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

## Acknowledgements
This work was part of the DOE Agile BioFoundry (http://agilebiofoundry.org), supported by the U. S. Department of Energy, Energy Efficiency and Renewable Energy, Bioenergy Technologies Office, and is based upon work supported by the DOE Joint BioEnergy Institute (http://www.jbei.org), U. S. Department of Energy, Office of Science, Biological and Environmental Research Program, under Award Number DE-AC02-05CH11231 between Lawrence Berkeley National Laboratory and the U.S. Department of Energy. This work was also supported by the Basque Government through the BERC 2022–2025 program and by the Spanish Ministry of Science and Innovation MICINN (AEI): BCAM Severo Ochoa excellence accreditation CEX2021-001142-S. Sandia National Laboratories is a multi-mission laboratory managed and operated by National Technology & Engineering Solutions of Sandia, LLC (NTESS), a wholly owned subsidiary of Honeywell International Inc., for the U.S. Department of Energy's National Nuclear Security Administration (DOE/NNSA) under contract DE-NA0003525. This written work is authored by an employee of NTESS. The employee, not NTESS, owns the right, title and interest in and to the written work and is responsible for its contents. The views and opinions of the authors expressed herein do not necessarily state or reflect those of the U.S. Government or any agency thereof. Neither the U.S. Government, nor any agency thereof, nor any of their employees makes any warranty, expressed or implied, or assumes any legal liability or responsibility for the accuracy, completeness, or usefulness of any information, apparatus, product, or process disclosed or represents that its use would not infringe privately owned rights. The U.S. Government retains and the publisher, by accepting the article for publication, acknowledges that the U.S. Government retains a nonexclusive, paid-up, irrevocable, worldwide license to publish or reproduce the published form of this manuscript, or allow others to do so, for U.S. Government purposes. The Department of Energy will provide public access to these results of federally sponsored research in accordance with the DOE Public Access Plan (http://energy.gov/downloads/doe-public-access-plan). Funding for open access charge: U.S. Department of Energy.

## Author contributions
Z.C., J.M.M., T.R., and H.G.M. conceived the original idea. Z.C., T.R., J.M.M., A.Z., M.I., V.B., M.S. M.T., A. P., C.E.L., S.T., T.O., N.K., T.B. developed the methodology. T.R., J.M.M., Z.C., A.Z., M.F., C.L., P.C.K., S.T., T.O., N.K. wrote the computer code. T.R., J.M.M., A.Z., P.C.K. performed numerical experiments. A.Z., M.R.I., T.R., J.M.M., C.E.L., T.B., M.T., A.P., T.E., V.B., T.C., S.T., T.O., N.K. performed physical experiments. A.Z., T.R., M.R.I., V.B. analyzed results. H.G.M., A.Z., M.R.I., A.M., N.J.H., J.D.K. wrote the paper.

## Competing interests
H.G.M. declares financial interests in Science AI Corporation. N.J.H. declares financial interests in TeselaGen Biotechnologies and in Ansa Biotechnologies. J.D.K. has financial interests in Ansa Biotechnologies, Apertor Pharma, Berkeley Yeast, BioMia, Cyklos Materials, Demetrix, Lygos, Napigen, ResVita Bio, and Zero Acre Farms.
