## [Transparent Peer Review file · Communications Biology]

Machine learning-led semi-automated medium optimization reveals salt as key for flaviolin production in *Pseudomonas putida*

Corresponding Author: Dr Hector Garcia Martin

Version 0:

Reviewer comments:

Reviewer #1

(Remarks to the Author)

Zournas et al presented an interesting approach to rationally guide the medium optimization to improve the biosynthesis of flaviolin. The authors presented an exemplified way of data availability, by making the complete protocol available online for download, which is much appreciated and will be beneficial for the community to take their approach for further applications. The topic of medium optimization is also of high interest, which plays an important but long-term overlooked role in developing new biotech processes. However, I have some critical points regarding the design of the combinations for the DBTL and the chosen of Abs340 as the indicator, which need to be addressed and clarified:

1. Sample size for the ML. The authors mentioned that the first two rounds of DBTL were conducted to generate the basement dataset for the ML-guided further DBTLs cycles. In this case, maximum 30 combinations could be tested, and if comparing this number to the total numbers if willing to optimize and balance 12 (or 13 components), the sample size is way too small to generate reliable dataset. This flow into two concerns:

i) the accuracy of the ML predictions. The authors also mentioned the unpredictable results from their predictions and the "random" performance between different DBTLs. Could this be possibly due to the limited sample size for the algorithm training?

ii) the starting design of 15 combinations of the medium which is very critical. Throughout the paper, I did not get exactly how this step was done, despite the authors mentioned the principle they used. But I guess it should also be done based on biological understandings to weight the relevance of different medium components to the biosynthesis and growth?

2. Connecting the design of the initial combination for optimization, I am wondering if it plays a role in the high NaCl demand, due to some algorithm effect? Maybe the authors gave it somewhere, but I will suggest the authors to give the "intermediate" medium recipe in the supplementary material before reaching the final optimum one. In this case, it will be easier to trace how the optimization was done, e.g. whether in the first cycle the NaCl concentration was already be increased significantly, etc.

3. Critical concerns of using 340 nm proxy. The figure S2 seems to give a linear correlation between the Abs340 and HPLC-determined concentration, but throughout the paper, we know this is not true. Particularly, if we look at the numbers at page 8 for instance, it says "The first campaign (C1, Fig. 4) generated media designs that improved titer by 148%, as measured by the proxy, and 70% when measured directly (through HPLC), starting at a production of ~95 mg/l flaviolin" and then "Campaign two (C2, Fig. S5) produced a higher improvement than C1, increasing titer by 170%, as measured by the proxy, and 60% when measured directly, starting at a production of ~95 mg/l flaviolin". If following the proxy as the standards, campaign 1 (148%) is weaker than campaign 2 (170%), and if using HPLC data, campaign 1 (70%) is better than the campaign 2 (60%). Such conflicts, together with other results, strongly suggested Abs340 cannot be used a reliable indicator for the production of flaviolin (at the least to the concentrations that the authors were working with). Since the whole paper is based on using Abs340 as the indicator to evaluate the optimizations, I think a major improvement will be needed here.

4. connecting the comment 3, I am not convinced why two indicators were used, both Abs340 and HPLC, and two (distinguished) percentage improvements were reported for each case. For me, only the reliable one should be used, while the other one should be discarded.

5. Connecting to the comment 4 above again, why was Abs340 selected, while the determination was done at Abs300 and Abs320 in HPLC?

Others

6. Introduction, paragraph 2, "10 components at 5 levels of concentration would require 50 experiment" is not valid. The

required experiment sizes will be magnitudes higher if aiming for an optimized medium recipe.

7. Novelty of the semi-automated pipeline for wet-lab screening. I am not quite convinced by the novelty of this approach. The sample size is only 15 combinations in 3 days, which is not superior compared to the current-status technology. It is based on the commercial BioLector system, and such a screening application at this capacity is kind of "standardized" level for this system.

8. It is too repetitive in terms of the text between the abstract and the last paragraph of the introduction.

9. How did the authors define the criteria to end their optimization, since 5 or 6 DBTLs were used in different cases? "No further improvements" seem not to be the one, as otherwise several cases would end up at DBTL 3 already.

10. The Abstract in the protocol has to be updated. It is inconsistent with the paper
[dx.doi.org/10.17504/protocols.io.81wgbx7eylpk/v1](https://doi.org/10.17504/protocols.io.81wgbx7eylpk/v1);

Reviewer #4

(Remarks to the Author)

Results

Related to the yields calculated by the authors, in the repositories, there are additional notebooks, such as `flavio_Max_Yield.ipnb`, which employ genome-scale metabolic models to calculate maximum yields. This approach is not mentioned in the main manuscript or supplementary data. Is there a reason for this omission? Including an analysis based on metabolic models could potentially enrich the study's findings. By simulating metabolic pathways and interactions, such an approach may help clarify underlying mechanisms and contextualize the experimental findings more comprehensively. For the explored trajectory analysis (Figure 5): The authors present PC1 and PC2, which together account for approximately 54% of the variance. However, it would be helpful to also include PC3, or other principal components, if they capture a significant percentage of variance. Adding supplementary figures that include these additional principal components could enhance the completeness of the results.

Minor changes:

- Figure 3: To reduce reader errors when comparing both heatmaps, I suggest adjusting the color scale to maintain consistency across both plots.
- Figure 4: Increase the font size of the axes labels and improve the overall figure quality.
- Figure S5: Increase the font size of the axes labels and enhance the figure quality. The plots are currently difficult to read.
- Figure S7: Similarly, increase the font size of the axes labels and improve the quality of the figure.

Code:

Following the authors' instructions, I encountered difficulties running the notebooks. For instance, the notebook `A_Find_Stock_Concentrations.ipynb` requires the `pyDOE` package. I suggest that the authors provide clearer setup instructions and include a complete list of requirements to facilitate code execution.

Reviewer #5

(Remarks to the Author)

The paper is expertly crafted, offering novel insights into the optimization of culture media and enhancing the titer and yield of flaviolin. Meanwhile, it shows that NaCl is the key factor to increase flaviolin production. Additionally, the implementation of a semi-automated pipeline and machine learning models significantly accelerates the Design-Build-Test-Learn (DBTL) cycles, thereby reducing manual labor. Nonetheless, there are a few errors that require meticulous correction. I would like to extend my appreciation to the author for their exceptional writing prowess. The language used is both clear and conversational, rendering the manuscript highly readable and straightforward to grasp.

Comments:

1. Figure 2 effectively conveys the comprehensive DBTL cycle concept of the paper. However, the icons representing Jupyter Notebook and Excel may not be the most suitable. It might be more appropriate to substitute these with visual representations of algorithms or related operational flowcharts. The style of the figure 2 is a little messy. Maybe you can make a simpler figure, or divide the figure into two pieces.
2. Figure 3 validates the reliability of four biological replicates, yet its inclusion in the main body of figures is subject to review. Please consider relocating this data to the supplementary materials for a more streamlined presentation.
3. In Figure 4, the red color used to denote the exploitation data in DBTL 3 is not consistent with the red color used in the corresponding data of DBTL 4 and DBTL 5. Is there a particular significance to this color variation? This is not addressed in the figure legend. If there is a specific reason for the discrepancy, please provide clarification; otherwise, it would be best to maintain color uniformity across the figures.
4. Could you provide additional insight into why a higher coefficient of determination (R^2) is desirable in the Figure 4? An expanded explanation of the role of R^2 in medium optimization and improving titer or yield would be beneficial.
5. For Figure 5, it might be worth considering a composite graph that combines the cycle number and Abs 340 data points, which would likely enhance the clarity and ease of interpretation. Besides, for Figure 6, renaming the x-axis to "Abs340 titer" could facilitate a more immediate comprehension by the audience.
6. Could you please describe in detail how your methods could be improved to accommodate more complex compound production? And in this paper, the microplate marker was used to measure the production of flaviolin. What do you think is the fastest way to get optimized data for the production of non-absorbent compounds using your protocol?

Version 1:

Reviewer comments:

Reviewer #1

(Remarks to the Author)

The authors have addressed my comments with mostly convincing arguments, and have made proper changes in their manuscript. I only have a few minor points and after these are to be addressed, I am happy to endorse the publication of this paper:

1. Table S1, n/a for the trace metal solution elements. This confuses me a bit, as i) the M9 medium should contain these ions in the recipe, and ii) if the compounds are not exist in the original recipe, how they become relevance in the further optimizations from C1-Cx.

2. concern of Abs340. I understand and also agree with the point that it is needed to have an easy measurable parameter for this type of screening test. I am still not quite convinced by the R2 of 0.74 between the Abs340 and the HPLC as a good correlation, also particularly the mismatch of the two parameters that I mentioned before. But I do agree that Abs340 is the best option so far for this work. What I will suggest the authors to add is: in the Material and Method section, please add one statement about the uncertainties of the Abs340, and make it clear that it is only a qualitative or at best semi-quantitative parameter. So the community is less likely to misuse it if one would like to follow this methodology in the future.

3. follow up to my comment 5 about the difference wavelength. Could the authors add the UV-Vis absorbance spectrum as supplementary material, to support the statement of "least amount of noise"?

Reviewer #4

(Remarks to the Author)

read the rebuttal and I saw the changes. I agree with them.

And related with the metabolic modelling, I understand that is out of scope but in my opinion, the GEMs integration can be a nice upgrade for future developments .

I was reading the ML discussion but I have no comments about it.

Reviewer #5

(Remarks to the Author)

The manuscript should be accepted as all my comments were solved. I have no other comments.

Reviewer #1 (Remarks to the Author):

Zournas et al presented an interesting approach to rationally guide the medium optimization to improve the biosynthesis of flaviolin. The authors presented an exemplified way of data availability, by making the complete protocol available online for download, which is much appreciated and will be beneficial for the community to take their approach for further applications. The topic of medium optimization is also of high interest, which plays an important but long-term overlooked role in developing new biotech processes.

We appreciate the positive comments of the reviewer, especially regarding the importance of media optimization and the data and protocol availability, both of which we hold in very high regard.

However, I have some critical points regarding the design of the combinations for the DBTL and the chosen of Abs340 as the indicator, which need to be addressed and clarified:

1. Sample size for the ML. The authors mentioned that the first two rounds of DBTL were conducted to generate the basement dataset for the ML-guided further DBTLs cycles. In this case, maximum 30 combinations could be tested, and if comparing this number to the total numbers if willing to optimize and balance 12 (or 13 components), the sample size is way too small to generate reliable dataset. This flow into two concerns:

i) the accuracy of the ML predictions. The authors also mentioned the unpredictable results from their predictions and the “random” performance between different DBTLs. Could this be possibly due to the limited sample size for the algorithm training?

The initial datasets involved 23 media designs for Campaign 1 and 31 media designs for Campaigns 2 and 3. While 23-31 media designs are not enough to provide accurate predictions for the full phase space involving 12-13 components, they are more than enough to start a successful active learning process that would eventually test 68-91 instances. The point of an active learning process is to use data collected from previous cycles to decide with media designs to test next, and improve production using the minimum amount of (expensive) experiments, as we have demonstrated in this work. The cross-validated R^2 after DBTL2 was ~ 0.3 in C1 and ~ 0.6 for C2 and C3 (see figure S5), which was more than enough to give us reliable recommendations for the following DBTL cycles. The fact that we could make use of only 23/30 media designs as a starting point for such a large phase space (12-13 components) and guide a successful active learning process is a benefit rather than a handicap, since experiments of this type are expensive to perform. We have modified the introduction to further expand the explanation of active learning:

“The significant cost of the large data sets needed to train ML algorithms prompts the use of active learning processes, in which ML algorithms select which data to collect. Active learning²⁹ uses ML in an iterative process in which the algorithm chooses the next set of experiments to be performed

(i.e., the next set of instances to be “labeled”). This approach increases data efficiency dramatically, minimizing the number of experiments that need to be performed to reach the desired goal (e.g., increase production). To date, we only know of two pioneering studies that use active learning to optimize media, and neither attempted to improve synthesis of a product.“

and the conclusion to stress the point that an active learning process can be very effective while minimizing the number of experiments:

“Finally, active learning can produce quite unexpected results if the phase space is made as wide as possible. However, a powerful predictive algorithm is required to efficiently search through this large phase space. In this case, the high predictive power and versatility of ART (Fig. 8), provided by the ensemble model design, and the quality of its recommendations, allowed us to explore a very large phase space with 12 (13 in campaign 3) variable components spanning 1-2 orders of magnitude each, even when starting from a very small training dataset (e.g. at DBTL3 of C1). This approach not only allowed us to optimize media while minimizing the number of (relatively expensive) experiments, but also to identify media designs that would not have been accessible using less powerful approaches, which might have required fully constraining some media components not expected to be important (e.g., NaCl).”

At no point do we talk about a “random performance” between different DBTL cycles. To the contrary, we have excellent reproducibility between DBTL cycles (Fig 3,4, S4), and between users (Fig. S3). We do mention non-deterministic paths converging to a similar region (Fig. 5), and a “bursty” behavior for the active learning process: i.e., response improvements come in bursts rather than as a smooth improvement when more data is added.

This is a general feature of active learning that has been reported in other recent papers:

Rapp, Jacob T., Bennett J. Bremer, and Philip A. Romero. "Self-driving laboratories to autonomously navigate the protein fitness landscape." *Nature chemical engineering* 1.1 (2024): 97-107.

Albuquerque, R. Q., Rothenhäusler, F. & Ruckdäschel, “H. Designing formulations of bio-based, multicomponent epoxy resin systems via machine learning.” *MRS Bull.* 49, 59–70 (2024).

and can also be observed in simulated active learning processes (see new supplementary Fig. S10). This is the reason we had to average over ten different processes to obtain Fig. 8. Hence, we believe the initial sample size plays no role in this “bursty” behavior.

We have modified the text to include these references and point towards the newly added figure:

“Rather, improvements in the response happened in bursts and in a rather unpredictable fashion (Fig. S10): they may happen when we have collected many instances or a few, and when our predictive power is either high or low. This behavior is in agreement with other active learning studies published recently^{68,69} “

ii) the starting design of 15 combinations of the medium which is very critical. Throughout the paper, I did not get exactly how this step was done, despite the authors mentioned the principle they used. But I guess it should also be done based on biological understandings to weight the relevance of different medium components to the biosynthesis and growth?

Each campaign was initiated by 23 (C1) or 31 (C2 and C3) media compositions selected by a traditional Design of Experiment (DoE) approach, Latin Hypercube Sampling (LHS), as we explain in the section “Applying the pipeline produced significant improvements in flaviolin production”. Latin Hypercube Sampling (LHS) is a Design of Experiment (DoE) algorithm that does not take into account any prior biological knowledge: LHS recommendations are created to span as much of the phase space as possible.

We have modified the manuscript to make it very clear that there was no biological prior knowledge used in the selection of initial media designs:

“All three campaigns were performed similarly: two DBTL cycles using DoE approaches, followed by three to four DBTL cycles of active learning guided by ART (**Fig. 4, Fig. S5**). DBTL cycles 1-2 were used to accumulate sufficient training data to make ART effective in predicting production from media composition. For this purpose, we used a DoE approach called Latin Hypercube Sampling (LHS), included in ART ⁴¹. LHS does not leverage any prior biological knowledge other than the components used and their upper and lower bounds, and is a purely statistical approach producing recommendations meant to span as much phase space as possible, since ML algorithms are typically much more effective for interpolating than extrapolating. After DBTL 1 and 2, an active learning process ensued, in which we used all the previously generated data from the prior DBTL cycles to train ART (e.g. training for DBTL 4 of C1 used the data from DBTL 1-3 of C1), and generated recommendations for the next DBTL cycle (**Fig. 2**).”

These 23-31 initial designs were then used to train a probabilistic model which provides the media designs for the subsequent cycle. This approach allows us to avoid selecting media composition based on biological understanding, other than the selection of the media components and their upper and lower bounds. We believe that this lack of biological bias constituted the main strength of our approach, and it allowed us to identify an optimal media composition with unexpectedly high concentration of NaCl. The effectiveness of the active learning pipeline can be seen by the increase in predictive power (R^2) as we progressed in DBTL cycles.

We have modified the text to make this point more evident:

“After DBTL 1 and 2, an active learning process ensued, in which we used all the previously generated data from the prior DBTL cycles to train ART (e.g. training for DBTL 4 of C1 used the data from DBTL 1-3 of C1), and generated recommendations for the next DBTL cycle (**Fig. 2**). These initial 23-31 datapoints were enough to generate effective recommendations. For example, in C1 the highest performing media designs was suggested in DBTL3, from a model trained only on 23 media designs. In

this active learning process we progressively shifted from an emphasis in exploration to exploitation. Explorative recommendations focused on investigating parts of the media phase space where ART's predictive power was most limited, whereas exploitative recommendations focused on suggesting new media designs that were predicted to yield the highest response⁴¹.”

2. Connecting the design of the initial combination for optimization, I am wondering if it plays a role in the high NaCl demand, due to some algorithm effect? Maybe the authors gave it somewhere, but I will suggest the authors to give the “intermediate” medium recipe in the supplementary material before reaching the final optimum one. In this case, it will be easier to trace how the optimization was done, e.g. whether in the first cycle the NaCl concentration was already be increased significantly, etc.

While all the media combinations tested here can be found in the github repositories, we agree with the reviewer that having this information in the text would be helpful, so we have included an intermediate media design in Supplementary Table S1. We are certain that the high importance of NaCl is not an artifact of the starting point or algorithm: the SHAP feature importance analysis shows that this is not the case, and the final results of all three campaigns very clearly indicate the effect that NaCl has on flaviolin production. For example, in Fig. 6B we show the NaCl concentrations we tested during C3, and we can see that there is a very substantial connection between performance of the media and NaCl concentration between 50mM to 450mM.

We modified the text in the following way to stress this point in the paper:

“Surprisingly, the salt NaCl emerged as the most critical feature overall influencing flaviolin production (**Fig. 7, S7, S8**). NaCl ranked first in feature importance for campaigns C1 and C2, and second for C3 (with glucose being the most important, as expected). In all three campaigns, the best-performing media contained 8-9.2 times the concentration in the starting media (400-460 mM NaCl, **Fig. S6**). **The consistency between the final results of three independent campaigns (Fig. 5), the feature importance analysis (Fig. 7), and the distinct effect of NaCl on flaviolin production (Fig. 6B) underscore the importance of NaCl for increased flaviolin production.** Similar, but much smaller, increases in titer by NaCl addition have been reported in other organisms and for diverse products. For example, NaCl has been shown to improve growth as well as isoprenol production in *E. coli* in the presence of ionic liquids

⁵⁴ “

3. Critical concerns of using 340 nm proxy. The figure S2 seems to give a linear correlation between the Abs340 and HPLC-determined concentration, but throughout the paper, we know this is not true. Particularly, if we look at the numbers at page 8 for instance, it says “The first campaign (C1, Fig. 4)) generated media designs that improved titer by 148%, as measured by the proxy, and 70% when measured directly (through HPLC), starting at a production of ~95 mg/l flaviolin” and then “Campaign two (C2, Fig. S5) produced a higher improvement than C1, increasing titer by 170%, as measured by the proxy, and 60% when measured directly, starting

at a production of ~95 mg/l flaviolin". If following the proxy as the standards, campaign 1 (148%) is weaker than campaign 2 (170%), and if using HPLC data, campaign 1 (70%) is better than the campaign 2 (60%). Such conflicts, together with other results, strongly suggested Abs₃₄₀ cannot be used a reliable indicator for the production of flaviolin (at the least to the concentrations that the authors were working with). Since the whole paper is based on using Abs₃₄₀ as the indicator to evaluate the optimizations, I think a major improvement will be needed here.

While we admit that the proxy Abs₃₄₀ measurement of flaviolin is not perfect, the main point of this paper is to show that an imperfect, but high-throughput, proxy can produce enough data to enable an active learning process that leads to improved production of flaviolin (as tested by the more reliable assay HPLC). Using HPLC to measure flaviolin production would not have allowed us to do as many DBTL cycles, and that would have impeded the performance of the ML algorithm, likely leading to worse overall flaviolin improvements. Abs₃₄₀ and HPLC measurements are linearly correlated ($R^2 = 0.74$, explaining 74% of the data variance), and that is sufficient for the active learning process to succeed. The fact that improvements in flaviolin production are different for each method is perfectly compatible with a good, but not perfect, correlation displaying an $R^2 = 0.74$. The noise involved in this proxy is one that the Automated Recommendation Tool is designed to handle (see Equation 2 in Radivojevic 2021). In the end, the important point is that flaviolin production was improved, and this was shown using HPLC, which is the most authoritative assay. Better proxies for flaviolin are desirable (e.g. Agilent RapidFire-MS), but are beyond this paper.

The case in which two measurements for a desired quantity are available, but one is quick and noisy and the other one is slow and more accurate is a common occurrence in biotechnology. Here we show how to productively leverage the quick and noisy assays, with results confirmed by the slow and more accurate assay.

We have added information about this in the conclusions of the paper:

"However, a powerful predictive algorithm is required to efficiently search through this large phase space. In this case, the high predictive power of ART, provided by the ensemble model design, and the quality of its recommendations, allowed us to explore a very large phase space with 12 (13 in campaign 3) variable components spanning 1-2 orders of magnitude each. This approach allowed us to identify media designs that would not have been accessible using less powerful approaches, which might have required fully constraining some media components not expected to be important (e.g., NaCl). In this study, optical measurements (Abs₃₄₀) correlated well with analytical measurements (HPLC), showing a coefficient of determination (R^2) equal to 0.74 (Fig. S2). Even though this correlation is not perfect, performing optical measurements allowed us to perform fast DBTL cycles, enabling rapid improvements in titer and yield, which were confirmed through the HPLC measurements (the golden standard). More accurate measurements might enable higher or faster improvements. Solutions like the Agilent RapidFire-MS or (ultra)-high-performance liquid chromatography (UHPLC) could bridge the gap between fast and accurate and fast measurements that enable machine learning in biology."

We also changed the caption in Fig. 4 to clear the distinction between actual titer (HPLC) and the titer proxy (Abs₃₄₀):

“The cross-validated predictions for the first two DBTL cycles showed limited predictive power ($R^2 = 0.32$). However, recommendations produced from this data improved the titer proxy (Abs₃₄₀) by 148% in DBTL 3.”

4. connecting the comment 3, I am not convinced why two indicators were used, both Abs340 and HPLC, and two (distinguished) percentage improvements were reported for each case. For me, only the reliable one should be used, while the other one should be discarded.

The reason we report both of these metrics is that we used Abs₃₄₀ as a high-throughput assay to guide the active learning process, and we used HPLC to validate the increases. In this way, we were able to produce a large enough amount of data for the machine learning algorithm to be effective, which would have been much more complicated using HPLC. The case in which two measurements for a desired quantity are available (one, quick and noisy and another one, slow and accurate) is common in biotechnology. Here we show how to productively leverage the quick and noisy assays, with results confirmed by the slow and more accurate assay (HPLC). For these reasons, we think it is important to report both metrics, even if Abs₃₄₀ is a less reliable metric since this is what our algorithm used to guide the media optimization process. This point is already in the text, and we have modified it to make it clearer:

"The microplate reader was used because our product, flaviolin, has light absorption properties that can be measured optically, accelerating phenotype acquisition with respect to other methods (HPLC, GC-MS, etc). In this way, we used the Abs₃₄₀ as a high-throughput assay to effectively guide the active learning process, and we used the HPLC assay to validate the increases with an authoritative assay."

5. Connecting to the comment 4 above again, why was Abs₃₄₀ selected, while the determination was done at Abs300 and Abs320 in HPLC?

We appreciate the attention to detail the reviewer has shown for our work. We selected Abs₃₄₀ as the metric since this wavelength was reported to show the least amount of noise, even if it is not the wavelength of maximal absorbance (see also comment 3). Flaviolin absorbs maximally at ~300nm, which we independently confirmed, but we selected 300nm for the HPLC because there is no interference with other biological molecules. In this way, we intended to maximize the signal to noise ratio. We neglected to add this information in the manuscript and we have corrected this error for this revision:

“In the end, media optimization in small wells is of limited use unless the results can be scaled to the higher volumes where production will take place. The microplate reader was used because our product, flaviolin, has light absorption properties that can be measured optically, accelerating phenotype acquisition with respect to other methods (HPLC, GC-MS, etc). In this way, we used the Abs₃₄₀ as a high-throughput assay to effectively guide the active learning process, and we used the HPLC assay to validate the increases with an authoritative assay. This approach has been reported previously in Yang *et*

al⁴⁵, where flavin was used as a Malonyl-CoA biosensor and the optimal wavelength for measurement was determined to be $\lambda = 340\text{nm}$, even though maximum absorbance is at $\lambda = \sim 300\text{nm}$. Final results were confirmed with HPLC (**Fig. S2**). To enable reproducibility through a standardized protocol description and transfer, the full protocol has been stored in protocols.io⁴⁵. “

To further clarify this point, we have also added a comment about this in the conclusions of the paper:

“The differences between Abs₃₄₀ and analytical quantification through HPLC indicate that, contrary to what has been previously reported⁴⁵, Abs₃₄₀ is not a very accurate proxy for flavin, especially when the cells are grown in significantly different physiological conditions. However, the linear correlation between the Abs₃₄₀ proxy and the HPLC (**Fig. S2**) shows that the use of Abs₃₄₀ can help identify better media designs. While using HPLC for all measurements would have resulted in more accurate measurements, it would have also slowed down the experimental work and it is not clear that we would have been able to identify media compositions that enabled almost double flavin production. Hence, a proxy, even if imperfect, can be more desirable than a more exact measurement if it accelerates the active learning process and the ML algorithm can manage the noise.”

Others

6. Introduction, paragraph 2, “10 components at 5 levels of concentration would require 50 experiment” is not valid. The required experiment sizes will be magnitudes higher if aiming for an optimized medium recipe.

That is absolutely correct: if we combinatorially tried to test all possible combinations for all the component concentrations then that would be the case. The point we were trying to make is that even testing them one component at a time would require 50 experiments. We have added the combinatorial calculation as well:

“This approach can be very time consuming for a typical media if testing all components: e.g, traditionally 10 components at 5 levels of concentration would require 50 experiments when tested one component at a time. Testing the combination of these would require 5¹⁰ experiments.”

7. Novelty of the semi-automated pipeline for wet-lab screening. I am not quite convinced by the novelty of this approach. The sample size is only 15 combinations in 3 days, which is not superior compared to the current-status technology. It is based on the commercial BioLector system, and such a screening application at this capacity is kind of “standardized” level for this system.

The novelty of this approach does not lie in the amount of data being produced, but rather in how the limited data being produced is very efficiently leveraged by ART to guide an active learning process that effectively improves production. Further, it does so in a systematic manner that can be applied to any product, pathway and host, and uses standard equipment (BioLector, Biomek) that is generally accessible. This constitutes a significant contribution to solve a problem that every bioengineer faces: media optimization.

We have added this point to end of the first paragraph of the conclusion:

“The concentration of NaCl had been previously known to affect production levels in other cases, but not shown to be the most important driver, and optimal values were not nearly as extreme as the one found in this study. We also showed, using synthetic data, that ART outperforms a widely used DoE approach (JMP) and other machine learning approaches (gpCAM) when leading the active learning process using synthetic data. ART requires fewer data points for similar outcomes or achieving better outcomes altogether (Fig. 8). These advantages are critical in cases where the cost of and time for data acquisition is high, such as in synthetic biology. The novelty of this approach resides not so much in the amount of data being produced, but rather in how the data being produced is very efficiently leveraged by ART to guide an active learning process that effectively improves production by optimizing media, a problem that every bioengineer faces.”

8. It is too repetitive in terms of the text between the abstract and the last paragraph of the introduction.

The last paragraph of the introduction gives an expanded description of the claims in the abstract, and for that reason we believe it is useful to the reader despite some redundancy. Hence, we have maintained the current wording.

9. How did the authors define the criteria to end their optimization, since 5 or 6 DBTLs were used in different cases? “No further improvements” seem not to be the one, as otherwise several cases would end up at DBTL 3 already.

Due to limited experimental bandwidth (experiments were costly) our goal was to run 5 DBTL cycles per campaign, unless we saw improvements in DBTL5, in which case we would proceed to one more cycle. Since we did not see that for DBTL5 in C1 and C3 we stopped at DBTL5, and since we saw an improvement at DBTL5 at C2, we proceeded to run an extra DBTL cycle to see if we could push performance further. Our goal was not so much to obtain the absolutely optimal composition, but to show that machine learning can provide unexpected insights into common processes.

We modified the text in the following way:

“As an example of this progressive shift from exploration to exploitation, in DBTL 3 from C2, 66% of the recommendations were explorative and 33% exploitative. In DBTL 4, however, 46% were explorative and 54% exploitative, and in DBTL 5, 33% were explorative and 66% exploitative (Table S2). Due to the time and monetary cost of biological experiments we limited the number of DBTL cycles performed in this study to numbers that are practical. Hence, we aimed to conduct 5 DBTL cycles per campaign. However, if we saw improvements at DBTL5, we would conduct further experiments until there was no further increase in the maximization objective for that campaign.”

10. The Abstract in the protocol has to be updated. It is inconsistent with the paper [dx.doi.org/10.17504/protocols.io.81wgbx7eylpk/v1](https://doi.org/10.17504/protocols.io.81wgbx7eylpk/v1);

We have matched the abstracts to be consistent. We have changed the main text in the following way, updating the links:

“In order to enhance reproducibility, the full protocol can be found in protocols.io ⁴⁶: [dx.doi.org/10.17504/protocols.io.81wgbx7eylpk/v2](https://doi.org/10.17504/protocols.io.81wgbx7eylpk/v2) includes the protocol to run the experiment and [dx.doi.org/10.17504/protocols.io.x54v9pr51g3e/v2](https://doi.org/10.17504/protocols.io.x54v9pr51g3e/v2) includes the protocol for measurement of Abs₃₄₀ and OD₆₀₀.”

Reviewer #4 (Remarks to the Author):

Results

Related to the yields calculated by the authors, in the repositories, there are additional notebooks, such as `flavio_Max_Yield.ipnb`, which employ genome-scale metabolic models to calculate maximum yields. This approach is not mentioned in the main manuscript or supplementary data. Is there a reason for this omission? Including an analysis based on metabolic models could potentially enrich the study's findings. By simulating metabolic pathways and interactions, such an approach may help clarify underlying mechanisms and contextualize the experimental findings more comprehensively.

Thanks for the proposal. However, the yield we reported in this study is the process yield (`flavolin_produced/glucose_added_in_the_media`), instead of the biological yield (`flavolin_produced/glucose_consumed_by_the_bacteria`). We cannot calculate the biological yield without measuring the glucose in the media after 48 hours, which we did not do. Since GEMs provide biological yield and we measured the process yield, we feel this comparison would be futile and confuse most readers, for very little gain. We have removed this notebook. This is a data-driven project using no prior biological knowledge, and we believe that is its strength. The use of GEMs, while interesting, is beyond the scope of this study.

For the explored trajectory analysis (Figure 5): The authors present PC1 and PC2, which together account for approximately 54% of the variance. However, it would be helpful to also include PC3, or other principal components, if they capture a significant percentage of variance. Adding supplementary figures that include these additional principal components could enhance the completeness of the results.

Thanks for the suggestion. We added a supplementary figure with a PCA including three principal components (Fig. S7).

Minor changes:

- Figure 3: To reduce reader errors when comparing both heatmaps, I suggest adjusting the color scale to maintain consistency across both plots.
- Figure 4: Increase the font size of the axes labels and improve the overall figure quality.

- Figure S5: Increase the font size of the axes labels and enhance the figure quality. The plots are currently difficult to read.
- Figure S7: Similarly, increase the font size of the axes labels and improve the quality of the figure.

We improved the quality of the figures according to the reviewer's suggestions.

Code:

Following the authors' instructions, I encountered difficulties running the notebooks. For instance, the notebook A_Find_Stock_Concentrations.ipynb requires the pyDOE package. I suggest that the authors provide clearer setup instructions and include a complete list of requirements to facilitate code execution.

In order to run the notebooks, an ART installation is required. We have now clarified this in the text:

Jupyter notebooks are available at https://github.com/JBEI/Flaviolin_media_opt_C1, https://github.com/JBEI/Flaviolin_media_opt_C2, https://github.com/JBEI/Flaviolin_media_opt_C3 (for campaigns C1, C2 and C3) and https://github.com/JBEI/media_compiler for the media compiler, under the LBNL open source license. In order to run the notebooks of each campaign, a functional ART installation is required. ART is free for academic use and a license can be requested as explained in <https://github.com/JBEI/ART#license>

Reviewer #5 (Remarks to the Author):

The paper is expertly crafted, offering novel insights into the optimization of culture media and enhancing the titer and yield of flaviolin. Meanwhile, it shows that NaCl is the key factor to increase flaviolin production. Additionally, the implementation of a semi-automated pipeline and machine learning models significantly accelerates the Design-Build-Test-Learn (DBTL) cycles, thereby reducing manual labor. Nonetheless, there are a few errors that require meticulous correction. I would like to extend my appreciation to the author for **their exceptional writing prowess**. The language used is both clear and conversational, **rendering the manuscript highly readable and straightforward to grasp**.

We thank the reviewer for their appreciation of our efforts to increase the readability of the manuscript.

Comments:

1. Figure 2 effectively conveys the comprehensive DBTL cycle concept of the paper. However, the icons representing Jupyter Notebook and Excel may not be the most suitable. It might be more appropriate to substitute these with visual representations of algorithms or related

operational flowcharts. The style of the figure 2 is a little messy. Maybe you can make a simpler figure, or divide the figure into two pieces.

In order to make it more understandable, we have divided figure 2 into two panels, and changed the caption accordingly. We have also eliminated the Jupyter and Excel icons.

2. Figure 3 validates the reliability of four biological replicates, yet its inclusion in the main body of figures is subject to review. Please consider relocating this data to the supplementary materials for a more streamlined presentation.

We believe that this figure is useful for two reasons: 1) it provides an intuitive understanding of the data collected and its variability 2) it directly showcases the good reproducibility obtained between replicates and cycles. For these reasons, and to show these points, we would like to keep it as Figure 3.

3. In Figure 4, the red color used to denote the exploitation data in DBTL 3 is not consistent with the red color used in the corresponding data of DBTL 4 and DBTL 5. Is there a particular significance to this color variation? This is not addressed in the figure legend. If there is a specific reason for the discrepancy, please provide clarification; otherwise, it would be best to maintain color uniformity across the figures.

Those colors were meant to be the same. We thank the reviewer for the attention to detail in this review. We have corrected this mistake.

4. Could you provide additional insight into why a higher coefficient of determination (R^2) is desirable in the Figure 4? An expanded explanation of the role of R^2 in medium optimization and improving titer or yield would be beneficial.

The coefficient of determination R^2 represents the fraction of the variance in the response variable (flaviolin production in this case) that can be predicted from the input variables (media component concentrations in this case): i.e., what percentage of the data is explained by the model. Hence, a higher value of R^2 implies a higher capability to predict flaviolin production from the media component concentrations, which is critical to find the media component concentrations that maximize production. We have added the following explanation to the “Applying the pipeline produced significant improvements in flaviolin production” introduction paragraph:

“We leveraged the semi-automated platform to perform three campaigns (C1, C2, C3, with 5-6 DBTL cycles each) and improve the titers and process yields of flaviolin production by 60% (148% as measured by Abs340), 70% (170% in Abs340) and 350% (300% in Abs340) respectively (Fig. 4, Fig. S5). The first two campaigns, C1 and C2, aimed to increase the flaviolin titer proxy (Abs340) while keeping the glucose concentration at the same level as our baseline media. For the third campaign, C3, we aimed to increase the process yield proxy (i.e., ratio of flaviolin proxy divided by the initial glucose concentration, see Materials and Methods) after unconstraining the glucose concentration. All three campaigns converged to

similar regions of the media phase space, even though the explored trajectory was different (Fig. 5). Indeed, the most successful media designs for each campaign were very similar, and displayed unexpectedly high levels of NaCl (close to the limits that *P. putida* can tolerate, Fig. S6). We evaluated the predictive accuracy of the model by using the coefficient of determination R^2 , which represents the fraction of the response data variance explained by the model (Chicco et al, 2021, Fig. 2B). A value close to one indicates very good predictions (almost all response data explained by the model), and values close to zero or negative indicate no predictive power. Hence, a higher value of R^2 is desirable because it implies a higher capability to predict flaviolin production from the media component concentrations, which is critical to find the media component concentrations that maximize production.”

And also included the following reference, which discusses at length the benefits of R^2 :

Chicco, Davide, Matthijs J. Warrens, and Giuseppe Jurman. "The coefficient of determination R-squared is more informative than SMAPE, MAE, MAPE, MSE and RMSE in regression analysis evaluation." Peerj computer science 7 (2021): e623

5. For Figure 5, it might be worth considering a composite graph that combines the cycle number and Abs 340 data points, which would likely enhance the clarity and ease of interpretation. Besides, for Figure 6, renaming the x-axis to "Abs340 titer" could facilitate a more immediate comprehension by the audience.

For Figure 5, we experimented with several options to generate a composite graph, but could not find a way to improve the clarity and ease of interpretation. For Figure 6, we changed the axes according to the reviewer comments.

6. Could you please describe in detail how your methods could be improved to accommodate more complex compound production? And in this paper, the microplate reader was used to measure the production of flaviolin. What do you think is the fastest way to get optimized data for the production of non-absorbent compounds using your protocol?

The optimization of production for more complex compounds than flaviolin can use exactly the same process as described here, with only a change in the compound assay. In order to provide enough DBTL cycles for the active learning to work effectively, it is desirable for this assay to be high-throughput. A biosensor is a possibility, as are faster analytical solutions such as the Agilent RapidFire-MS or (ultra)-high-performance liquid chromatography (UHPLC). We have included these considerations in the conclusion of the paper:

“However, a powerful predictive algorithm is required to efficiently search through this large phase space. In this case, the high predictive power of ART, provided by the ensemble model design, and the quality of its recommendations, allowed us to explore a very large phase space with 12 (13 in campaign 3) variable components spanning 1-2 orders of magnitude each. This approach allowed us to identify media designs that would not have been accessible using less powerful approaches, which might have required fully constraining some media components not expected to be important (e.g., NaCl). In this study, optical measurements (Abs_{340}) correlated well with analytical measurements (HPLC), showing a coefficient of

determination (R^2) equal to 0.74 (**Fig. S2**). Even though this correlation is not perfect, performing optical measurements allowed us to perform fast DBTL cycles, enabling rapid improvements in titer and yield, which were confirmed through the HPLC measurements (the golden standard). More accurate measurements might enable higher or faster improvements. Solutions like the Agilent RapidFire-MS or (ultra)-high-performance liquid chromatography (UHPLC) could bridge the gap between fast and accurate and fast measurements that enable machine learning in biology.”

Reviewer #1 (Remarks to the Author):

REVIEWERS' COMMENTS:

Reviewer #1 (Remarks to the Author):

The authors have addressed my comments with mostly convincing arguments, and have made proper changes in their manuscript. I only have a few minor points and after these are to be addressed, I am happy to endorse the publication of this paper:

1. Table S1, n/a for the trace metal solution elements. This confuses me a bit, as i) the M9 medium should contain these ions in the recipe, and ii) if the compounds are not exist in the original recipe, how they become relevance in the further optimizations from C1-Cx.

Not all the ions included in the MOPS minimal media are also present in the recipe for the M9 media. We did not include components that are in the M9 but not in MOPS minimal media, since the numbers shown in the table are only for reference. Since A5 trace minerals are sometimes included in the M9 recipe, we have included these (H_3BO_3 , $(\text{NH}_4)_2\text{Mo}_7\text{O}_{22}$, CuSO_4 , ZnSO_4) in the table to avoid confusion.

2. concern of Abs₃₄₀. I understand and also agree with the point that it is needed to have an easy measurable parameter for this type of screening test. I am still not quite convinced by the R² of 0.74 between the Abs₃₄₀ and the HPLC as a good correlation, also particularly the mismatch of the two parameters that I mentioned before. But I do agree that Abs₃₄₀ is the best option so far for this work. What I will suggest the authors to add is: in the Material and Method section, please add one statement about the uncertainties of the Abs₃₄₀, and make it clear that it is only a qualitative or at best semi-quantitative parameter. So the community is less likely to misuse it if one would like to follow this methodology in the future.

We have added the following statement to the methods:

This is likely the result of differences in the limits of detection between the two measurements. Indeed, the good but not too high correlation between Abs₃₄₀ in our culture supernatant and HPLC measurements ($R^2 = 0.74$), indicate that Abs₃₄₀ is, at best, a semi-quantitative proxy for flavin. This is especially true when cells are grown in very diverse physiological conditions, since there is background Abs₃₄₀, (Fig. S1, Yang et al. 2018)

3. follow up to my comment 5 about the difference wavelength. Could the authors add the UV-Vis absorbance spectrum as supplementary material, to support the statement of “least amount of noise”?

We cannot add the UV-Vis absorbance spectrum as supplementary material because of copyright issues: we leveraged the one measured by Yang et al. (we did not perform the same experiments as they did). However, we have cited in the methods the Figure S1 of that study, which has the requested UV-Vis absorbance spectrum:

This is especially true when cells are grown in very diverse physiological conditions, since there is background Abs₃₄₀, (Fig. S1, (Yang et al. 2018))

Yang, D. *et al.* Repurposing type III polyketide synthase as a malonyl-CoA biosensor for metabolic engineering in bacteria. *Proc Natl Acad Sci USA* **115**, 9835–9844 (2018).

Reviewer #4 (Remarks to the Author):

I read the rebuttal and I saw the changes. I agree with them.
And related with the metabolic modelling, I understand that is out of scope but in my opinion, the GEMs integration can be a nice upgrade for future developments .
I was reading the ML discussion but I have no comments about it.

Reviewer #5 (Remarks to the Author):

The manuscript should be accepted as all my comments were solved. I have no other comments.